# Flavoured $(g-2)_\mu$ with dark lepton seasoning

Harun Acaroğlu[1,2], Prateek Agrawal[2] and Monika Blanke[1,3]

**1** Institut für Theoretische Teilchenphysik, Karlsruhe Institute of Technology,
Engesserstraße 7, D-76128 Karlsruhe, Germany
**2** Rudolf Peierls Centre for Theoretical Physics, University of Oxford,
Parks Road, Oxford OX1 3PU, United Kingdom
**3** Institut für Astroteilchenphysik, Karlsruhe Institute of Technology,
Hermann-von-Helmholtz-Platz 1, D-76344 Eggenstein-Leopoldshafen, Germany

## Abstract

As a joint explanation for the dark matter (DM) problem and the muon $(g-2)$ anomaly, we propose a simplified model of lepton-flavoured complex scalar DM with couplings to both the left- and right-handed leptons of the Standard Model (SM). Both interactions are governed by the same new flavour-violating coupling matrix $\lambda$, however we allow for a relative scaling of the coupling strength. The SM is further extended by two fermion representations, transforming as an $SU(2)_L$ doublet and singlet, respectively, and mediating these interactions. The fermions additionally couple to the SM Higgs doublet via a new Yukawa coupling. To study the model's phenomenology we first investigate constraints from collider searches, flavour experiments, precision tests of the SM, the DM relic density, and direct as well as indirect detection experiments individually. We then perform a combined analysis by demanding that all mentioned constraints are satisfied simultaneously. We use the results of this combined analysis and examine if the model is capable of accommodating the $(g-2)_\mu$ anomaly within its viable parameter space without introducing fine-tuned lepton masses. For all benchmark scenarios we consider, we find that the central value of $\Delta a_\mu^{\text{exp}}$ can be reached without generating too large corrections to the lepton masses. We hence conclude that this model qualifies as a viable and attractive lepton-flavoured DM model that at the same time solves the $(g-2)_\mu$ anomaly.


doi:[10.21468/SciPostPhys.15.4.176]()

# 1  Introduction

Within the wide range of different DM candidates [1], those predicted by models which are capable of addressing other problems, puzzles or anomalies of physics appear particularly appealing. Among the latter, the discrepancy between experimental measurements [2] and state of the art SM predictions [3] of the muon anomalous magnetic dipole moment $a_\mu$ currently constitutes one of the most significant hints at new physics (NP) and is referred to as the muon $(g-2)$ anomaly. In order to reconcile solutions to these two problems, models of DM coupling to the leptonic sector of the SM are worth investigating. In fact, flavoured dark matter (FDM) models, coupling to either SM quarks or leptons, have proven to generally exhibit a rich phenomenology and have been the subject of many previous studies [4–13].

In FDM models the DM field – as the name already suggests – carries flavour and thus comes in multiple (usually three) generations. As this potentially also allows for strongly constrained new sources of flavour violation, early studies of such models have been restricted to the Minimal Flavour Violation (MFV) framework [14–19] in which all new flavour violating

interactions are expressed in terms of the SM Yukawa couplings. As this approach yields a highly constrained flavour structure for the coupling matrix $\lambda$ that governs the interactions between DM and the SM, more general studies [20–26] have been performed within the Dark Minimal Flavour Violation (DMFV) framework presented in [20]. This framework allows for a generic flavour structure of $\lambda$ by proposing that it constitutes the only new source of flavour violation besides the SM Yukawa couplings.

Within the context of the muon $(g-2)$ anomaly, lepton-flavoured DM models [27–30] are of particular interest, since they assume the DM field to couple to leptons and can therefore generate potentially sizable contributions to $(g-2)_\mu$. Thus, the main subject of the present study is a lepton-flavoured DM model which extends the model presented in our previous paper [26]. There we had studied a lepton-flavoured complex scalar DM model within the DMFV framework in which DM coupled to the right-handed charged leptons through the exchange of a charged vector-like fermion. While that model has a rich and interesting phenomenology, due to the chiral structure of the model, we found the new contributions to the muon anomalous magnetic moment to be negligible. Hence, in this paper we extend the model from [26] by an additional mediator field and its interactions: the field content is extended by a vectorlike fermionic $SU(2)_L$ doublet containing one neutral and one charged state which couple DM to the SM left-handed lepton doublets. We further couple this new fermionic doublet and the charged fermion singlet mediating the DM interactions with the right-handed charged leptons to the SM Higgs doublet through a new Yukawa coupling $y_\psi$.[1] This model does not belong to the DMFV class as its coupling structure is inconsistent with the DMFV flavour symmetry assumptions, due to the additional left-handed interactions. Yet, in order to keep the number of new coupling parameters manageable, we assume the interaction between DM and left-handed leptons to be governed by the same coupling matrix $\lambda$ as the right-handed interactions. However, we allow for a scaling of this coupling in terms of a parameter $\xi$ in order to overcome the ad-hoc nature of choosing the couplings of right- and left-handed interactions to be equal.

We start our analysis by introducing the details of the model described above and especially presenting its mass spectrum. We then study its phenomenology by subsequently analysing constraints from collider searches, flavour experiments, precision tests of the SM, the DM relic density, and direct as well as indirect DM detection experiments. Subsequently, we perform a combined analysis in which we demand that all constraints are satisfied simultaneously in order to determine the viable parameter space of the model. Finally we examine if this model can generate sizeable contributions to the muon anomalous magnetic moment $a_\mu$, keeping an eye on potentially large accompanying corrections to the muon mass that could introduce fine-tuning and hereby render this solution to the $(g-2)_\mu$ anomaly unattractive.

## 2 Model setup

We use this section to present our simplified lepton-flavoured DM model coupling to both left- and right-handed leptons, pointing out important differences to DMFV models and discussing its mass spectrum in particular.

### 2.1 Lepton-flavoured DM with left- and right-handed couplings

We propose a simplified model which extends the SM by three complex scalar fields and two fermion representations. The scalar fields are contained in the dark flavour triplet

---

[1] A solution to the $(g-2)_\mu$ anomaly with similar field content, but without the flavour structure and DM interpretation, has previously been investigated in [31, 32].

Table 1: NP fields and their definitions as well as their representations under the SM gauge group.

| Field | Definition | $SU(3)_C$ | $SU(2)_L$ | $U(1)_Y$ | Spin |
|---|---|---|---|---|---|
| $\phi$ | $(\phi_1, \phi_2, \phi_3)^T$ | $\mathbf{1}$ | $\mathbf{1}$ | $0$ | $0$ |
| $\Psi$ | $(\psi_0, \psi'_1)^T$ | $\mathbf{1}$ | $\mathbf{2}$ | $-1/2$ | $1/2$ |
| $\psi'_2$ | $-$ | $\mathbf{1}$ | $\mathbf{1}$ | $-1$ | $1/2$ |

$\phi = (\phi_1, \phi_2, \phi_3)^T$ and have quantum numbers $(\mathbf{1}, \mathbf{1}, 0)_0$, where we use the short-hand notation $(SU(3)_c, SU(2)_L, U(1)_Y)_{\text{spin}}$. They couple to the SM's left- and right-handed lepton fields through the doublet $\Psi = (\psi_0, \psi'_1)^T$ with quantum numbers $(\mathbf{1}, \mathbf{2}, -1/2)_{1/2}$ and the singlet $\psi'_2$ which transforms as $(\mathbf{1}, \mathbf{1}, -1)_{1/2}$, respectively. The two new fermion fields are additionally Yukawa-coupled to the SM Higgs doublet $H$. We assume that the lightest generation of $\phi$ constitutes the observed DM in the universe. An overview of the NP fields and their representations under the SM gauge group is given in Table 1. We further assume the new fields $\phi$, $\Psi$ and $\psi'_2$ to be charged under a discrete $\mathbb{Z}_2$ symmetry. The Lagrangian of this model reads

$$
\begin{aligned}
\mathcal{L} = {} & \mathcal{L}_{\text{SM}} + \bar{\Psi}(i\slashed{D} - m_\Psi)\Psi + \bar{\psi}'_2(i\slashed{D} - m_\psi)\psi'_2 + (\partial_\mu \phi)^\dagger (\partial^\mu \phi) - \phi^\dagger M_\phi^2 \phi \\
& - (\lambda_{ij}^R \, \bar{\ell}_{Ri}\psi'_2 \, \phi_j + \lambda_{ij}^L \, \bar{L}_i \Psi \, \phi_j + y_\psi \, \bar{\Psi}\psi'_2 H + \text{h.c.}) \\
& + \lambda_{H\phi} \, \phi^\dagger \phi \, H^\dagger H + \lambda_{\phi\phi} \left(\phi^\dagger \phi\right)^2 .
\end{aligned}
\tag{1}
$$

Here, the couplings $\lambda^{R/L}$ are complex $3 \times 3$ matrices. In order to keep the total number of free parameters manageable, we assume that the left-handed coupling $\lambda^L$ is related to $\lambda^R$ through

$$
\lambda^L = \xi \lambda^R = \xi \lambda ,
\tag{2}
$$

i.e. left- and right-handed couplings are equal up to a scaling parameter $\xi$. In this way we ensure that the NP couplings to the SM lepton sector are governed by a single new flavour-violating matrix $\lambda$. We note that, while similar to the DMFV Ansatz, this simplifying assumption can not be traced back to a new flavour symmetry. To overcome its rather ad-hoc nature and to ensure that the entirety of the model's phenomenology is captured in our analysis, we allow $\xi$ to be a complex number such that effects due to a relative phase between $\lambda^R$ and $\lambda^L$ can still be present. Note that the scaling parameter $\xi$ is particularly relevant in Section 4, where we discuss constraints from lepton flavour violating (LFV) decays, as large contributions from diagrams with a chirality flip in the loop can be suppressed through $\xi$. At the same time $\xi$ should not suppress left-handed interactions too strongly, as equivalent chirality-flipping contributions to the muon anomalous magnetic moment $a_\mu$ are needed in order to generate sizeable NP effects within the mass ranges allowed by collider searches. We provide a detailed discussion of this interplay of different constraints and their impact on the scaling parameter $\xi$ in our phenomenological analysis.

The mass parameters $m_\Psi$ and $m_\psi$ as well as the mass matrix $M_\phi$ are discussed in detail in Section 2.2. While the coupling $\lambda_{\phi\phi}$ is only given for completeness here and has no impact on our analysis, the Higgs portal coupling $\lambda_{H\phi}$ actually bears relevance that we comment on whenever necessary.

Contrary to the models studied in [20–25] and especially in [26], where the DM triplet had purely right-handed interactions with the SM fermion sector, in the present model, there is no flavour symmetry that the DM flavour triplet $\phi$ can be associated with, due to its couplings

to both left- and right-handed leptons in eq. (1). Thus, all parameters of the matrix $\lambda$ remain physical, and we write it in terms of nine real parameters and nine complex phases, i.e.

$$\lambda_{ij} = |\lambda_{ij}| e^{i\delta_{ij}}. \tag{3}$$

This yields a total number of 18 physical parameters that the coupling matrix $\lambda$ depends on, which together with the mass parameters $m_\Psi$, $m_\psi$ and the three $m_{\phi_i}$ as well as the Yukawa coupling $y_\psi$ and the scaling parameter $\xi$ amounts to a total number of 26 free parameters. To ensure perturbativity and avoid a double-counting of the parameter space we will scan over the ranges

$$|\lambda_{ij}| \in [0,2], \quad \delta_{ij} \in [0,2\pi), \quad y_\psi \in [0,2], \quad |\xi| \in (0,1], \quad \delta_\xi \in [0,2\pi), \tag{4}$$

in our phenomenological analysis. Note that the Yukawa coupling $y_\psi$ can be taken real and non-negative without loss of generality. We restrict the absolute value of $\xi$ to be smaller than one, since we consider this model as an extension of the one studied in Reference [26] to reproduce the experimental value of the muon anomalous magnetic moment $(g-2)_\mu$. Therefore we assume the right-handed lepton coupling to be dominant, i.e. $|\xi| \leq 1$.

## 2.2 Mass spectrum and DM stability

The Yukawa interaction between $\Psi$, $\psi'_2$ and the Higgs doublet $H$ in eq. (1) introduces a mixing of the charged fermions $\psi'_1$ and $\psi'_2$ with the corresponding mass matrix

$$M_\psi = \begin{pmatrix} m_\Psi & \frac{v\,y_\psi}{\sqrt{2}} \\ \frac{v\,y_\psi}{\sqrt{2}} & m_\psi \end{pmatrix}, \tag{5}$$

where $v = 246\,\text{GeV}$ is the vacuum expectation value of the Higgs field. Using the Ansatz

$$\begin{pmatrix} \psi'_1 \\ \psi'_2 \end{pmatrix} = \begin{pmatrix} \cos\theta_\psi & -\sin\theta_\psi \\ \sin\theta_\psi & \cos\theta_\psi \end{pmatrix} \begin{pmatrix} \psi_1 \\ \psi_2 \end{pmatrix}, \tag{6}$$

we diagonalize this mass matrix to find the eigenvalues

$$m_{\psi_1} = \frac{1}{2}\left(m_\Psi + m_\psi + \sqrt{(m_\Psi - m_\psi)^2 + 2\,y_\psi^2\,v^2}\right), \tag{7}$$

$$m_{\psi_2} = \frac{1}{2}\left(m_\Psi + m_\psi - \sqrt{(m_\Psi - m_\psi)^2 + 2\,y_\psi^2\,v^2}\right), \tag{8}$$

with the corresponding mixing angle

$$\theta_\psi = \frac{1}{2}\arccos\left(\frac{(m_\Psi - m_\psi)}{\sqrt{(m_\Psi - m_\psi)^2 + 2\,y_\psi^2\,v^2}}\right). \tag{9}$$

We can then write the Lagrangian from eq. (1) in terms of the mass eigenstates $\psi_1$ and $\psi_2$ and find

$$\begin{aligned}
\mathcal{L} = {}& \mathcal{L}_{\text{SM}} + \bar{\psi}_0(i\slashed{D} - m_{\psi_0})\psi_0 + \bar{\psi}_1(i\slashed{D} - m_{\psi_1})\psi_1 + \bar{\psi}_2(i\slashed{D} - m_{\psi_2})\psi_2 + (\partial_\mu\phi)^\dagger(\partial^\mu\phi) \\
& - \phi^\dagger M_\phi^2 \phi + \lambda_{H\phi}\,\phi^\dagger\phi\,H^\dagger H + \lambda_{\phi\phi}\left(\phi^\dagger\phi\right)^2 - \left\{\xi\lambda_{ij}\,\bar{\nu}_i P_R \psi_0 \phi_j + \text{h.c.}\right\} \\
& - \left\{\lambda_{ij}\bar{\ell}_i\left[\left(\cos\theta_\psi P_L - \xi\sin\theta_\psi P_R\right)\psi_2 + \left(\sin\theta_\psi P_L + \xi\cos\theta_\psi P_R\right)\psi_1\right]\phi_j + \text{h.c.}\right\} \\
& - \frac{y_\psi}{\sqrt{2}}\left\{\sin 2\theta_\psi\left[\bar{\psi}_1\psi_1 - \bar{\psi}_2\psi_2\right]h + \cos 2\theta_\psi\left[\bar{\psi}_1\psi_2 + \bar{\psi}_2\psi_1\right]h\right\},
\end{aligned} \tag{10}$$

where we additionally defined $m_{\psi_0} = m_\Psi$. The absence of a global flavour symmetry also has implications for the DM mass matrix $M_\phi^2$, as it cannot be parametrized by $\lambda$ through the usual DMFV spurion expansion [20]. We thus choose $M_\phi^2$ to be diagonal by writing

$$M_\phi^2 = \text{diag}(m_{\phi_1}^2, m_{\phi_2}^2, m_{\phi_3}^2), \tag{11}$$

which we complement by the conventional hierarchy $m_{\phi_1} > m_{\phi_2} > m_{\phi_3}$.[2] Note that in contrast to the models presented in [20–26], here the masses $m_{\phi_i}$ are free parameters[3] which in turn means that the mass splittings between different dark flavours are not restricted. However, to keep the results of our analysis comparable to the studies performed in [20–26] we restrict them to a maximum of 30%. To ensure that the lightest state $\phi_3$ is stable we additionally impose a $\mathbb{Z}_2$ symmetry under which only the new fields $\phi_i$ and $\psi_\alpha$[4] are charged, such that their decays into SM-only final states are forbidden. This guarantees that $\phi_3$ is stable as long as it constitutes the lightest NP state. We further choose to work with the convention $m_{\psi_1} \geq m_{\psi_2}$. Since the mixing between the two charged mediators $\psi_{1,2}$ implies that one of them is heavier than the neutral state $\psi_0$ while the other is lighter, we thus always have the hierarchy

$$m_{\psi_1} \geq m_{\psi_0} \geq m_{\psi_2} > m_{\phi_3}. \tag{12}$$

Recall that $\psi_0$ is neutral while $\psi_{1,2}$ carry electric charge -1. We further work in the limit of zero neutrino masses $m_{\nu_i}$ throughout this analysis, which holds to an excellent approximation.

## 3 Collider phenomenology

Collider searches place important constraints on the mass parameters of the new particles $\psi_\alpha$ and $\phi_i$. We use this section to discuss the implications of LHC searches for the parameter space of our model. To reconcile our analysis with results from the LEP experiments [33,34] we assume that the charged mediators are heavier than 100 GeV, i.e. we choose $m_{\psi_{1/2}} > 100$ GeV.

### 3.1 Relevant LHC signatures

The annihilation of a quark and an antiquark from the initial state protons into an off-shell electroweak boson or photon gives rise to the production of mediator pairs $\bar{\psi}_\alpha \psi_\beta$ with $\alpha, \beta \in \{0,1,2\}$. This is shown in Figure 1. Here the indices depend on the off-shell $s$-channel boson—mixed pairs of $\psi_0$ and either $\psi_1$ or $\psi_2$ can only be produced if the Drell-Yan process is mediated by a $W$ boson, while the production of mixed pairs of $\psi_1$ and $\psi_2$ is mediated by $h$ or $Z$. Also, only a $Z$ boson can decay into the state with a $\psi_0$ pair.

The subsequent decay of the fermions $\psi_\alpha$ shown in Figure 2 then gives rise to several signatures depending on the constellation of the intermediate state explained above. While the charged mediators $\psi_{1,2}$ decay into a charged lepton and a dark scalar, the neutral mediator $\psi_0$ decays into a neutrino and a dark scalar and leaves no trace in the detector. We thus obtain mono- as well as di-lepton signatures in association with missing transverse energy. Other possible signatures arise from cascade decays of the heavier mediators $\psi_\alpha$ into lighter mediators $\psi_\beta$ and a $W$, $Z$ or Higgs boson. The subsequent decay of the gauge boson into

---

[2]In general, $M_\phi^2$ receives a correction from the Higgs portal coupling $\lambda_{H\phi}$ after EW symmetry breaking. Since we assume $\lambda_{H\phi}$ to be negligible throughout our analysis, we omit this term here.

[3]We assume the masses $m_{\phi_i}$ to include radiative corrections, i.e. we take them to be the renormalised on-shell masses.

[4]We use Greek indices when generally referring to any of the mass eigenstates $\psi_0$, $\psi_1$ and $\psi_2$ throughout this analysis. This should not be confused with the usual convention of using Greek letters as spinor indices.

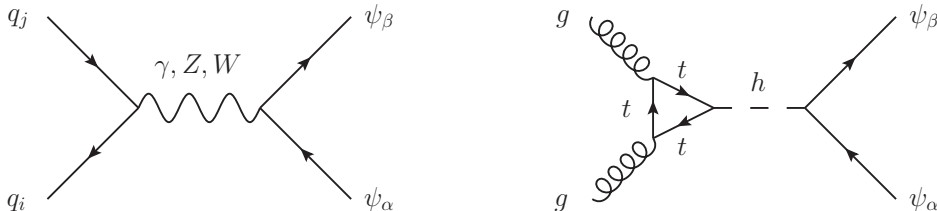

Figure 1: Feynman diagrams for the production of $\bar{\psi}_\alpha \psi_\beta$ pairs through a Drell-Yan process mediated by electroweak gauge bosons or by a Higgs boson produced by gluon fusion.

leptons and the lighter mediator's decay into a lepton and a dark scalar then give rise to signatures with three or more charged leptons and missing energy. Collecting all these decay channels, we find the following relevant processes for LHC searches:

$$
\begin{aligned}
pp &\rightarrow \bar{\psi}_0 \psi_\alpha \rightarrow \bar{\nu}_i \ell_j \phi_k^\dagger \phi_l, \\
pp &\rightarrow \bar{\psi}_\alpha \psi_\beta \rightarrow \ell_i \bar{\ell}_j \phi_k^\dagger \phi_l, \\
pp &\rightarrow \bar{\psi}_0 \psi_1 \rightarrow \bar{\nu}_i \bar{\ell}_j \ell_j \ell_k \phi_l^\dagger \phi_m, \\
pp &\rightarrow \bar{\psi}_0 \psi_2 \rightarrow \bar{\ell}_j \bar{\nu}_i \ell_i \ell_k \phi_l^\dagger \phi_m,
\end{aligned}
\tag{13}
$$

where $\alpha, \beta \in \{1, 2\}$ and $i, j, k, l$ and $m$ are flavour indices. Here we have omitted charge conjugated processes and final states with more than three leptons for brevity. In total these processes yield the signatures $\ell_i + \not{E}_T$, $\ell_i \bar{\ell}_j + \not{E}_T$, $\bar{\ell}_i \ell_i \ell_j + \not{E}_T$ and $\bar{\ell}_i \ell_j \ell_k + \not{E}_T$.

Since existing searches for the mono-lepton signature [35, 36] consider NP cases with different kinematics, a proper recasting would be in place in order to derive meaningful constraints on our model's parameter space. Additionally, the mono-lepton signature suffers from a large SM background stemming from $s$-channel $W$ production with subsequent decay into a charged lepton and a neutrino. We thus expect this signature to yield subleading constraints and therefore ignore it in our analysis. The signatures $\bar{\ell}_i \ell_i \ell_j + \not{E}_T$ and $\bar{\ell}_i \ell_j \ell_k + \not{E}_T$ only differ by the case with $i \neq j \neq k$, i.e. the case with an electron, a muon and a tau in the final state. The latter final states are correlated with the strongly constrained LFV decays in many models, such as supersymmetry, and therefore no dedicated LHC searches are available. Existing searches for the signature $\bar{\ell}_i \ell_i \ell_j + \not{E}_T$ again exhibit different final state kinematics [37], such that a thorough recasting is necessary in order to derive applicable constraints for our model, which we leave for future work. We thus focus on the di-lepton+$\not{E}_T$ signature in this work.

Lastly, we neglect mixed-flavour final states with $i \neq j$ in this analysis although, in contrast to non-flavoured DM models, these signatures do not require flavour violation in the coupling matrix that governs the interaction between DM and the SM in our model. In doing so, we follow the results of our analysis in [26] stating that searches in same-flavour final states already exclude the region of the parameter space in which the mixed-flavour final states yield rates comparable to the SM background. This leaves us with the signatures $e\bar{e} + \not{E}_T$, $\mu\bar{\mu} + \not{E}_T$ and $\tau\bar{\tau} + \not{E}_T$. As we showed in [26], searches for final states with a pair of taus [38] can be neglected as well, since they yield significantly weaker limits than searches for final states with light leptons. Those final states are constrained by searches for supersymmetric scalar leptons (sleptons) of the first and second generation, which in SUSY models are pair-produced and subsequently decay into a neutralino and a charged lepton. This leads to the relevant signature $\ell\bar{\ell} + \not{E}_T$ with $\ell = e, \mu$, where in the experimental analyses $\mu - e$ universality is commonly assumed.

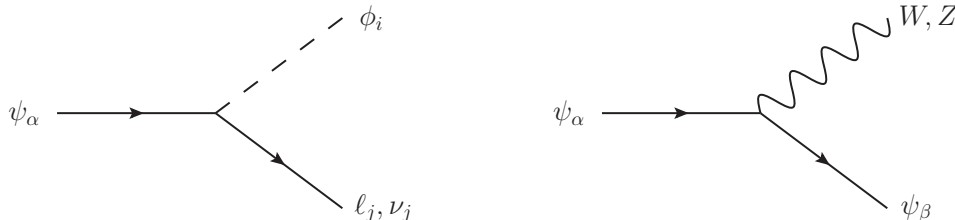

Figure 2: Feynman diagrams for the decay of $\psi_\alpha$ into leptons and dark matter (left) and gauge bosons and lighter mediators $\psi_\beta$ with $m_{\psi_\beta} < m_{\psi_\alpha}$ (right). For the latter we only show decays into electroweak gauge bosons and $\psi_\beta$ while decays into a Higgs boson and $\psi_\beta$ are possible as well.

## 3.2 Recast of LHC limits

Amongst several experimental searches for sleptons in final states with two charged leptons and missing transverse energy [39–42] the CMS search in [39] places the strongest constraints on the parameter space of our model. This search uses the full run 2 data set with an integrated luminosity of $137\,\text{fb}^{-1}$. In order to properly recast the bounds from [39] which we obtained from the SModelS [43] database, we implement the Lagrangian from eq. (1) in FeynRules [44]. Using this implementation we generate a UFO file [45] and calculate the leading-order signal cross section of the relevant process in MadGraph 5 [46]. To constrain the parameter space of our model we then compare the signal cross section to the experimental upper limit obtained from the above mentioned searches. In doing this, we neglect the impact of the potentially different final-state kinematics due to the different spin-statistics in our model relative to the SUSY case.

In our numerical analysis of the LHC constraints we follow [20–26] and ignore possible mass splittings between the different dark flavours $m_{\phi_i}$ discussed in Section 2.2, as small splittings only lead to additional soft and therefore difficult-to-detect decay products. We further neglect flavour-violating effects and consider a diagonal coupling matrix $\lambda$. Allowing for flavour-violating effects, i.e. off-diagonal elements in $\lambda$ would reduce the branching ratio of a given flavour-conserving final state and therefore reduce its signal cross section. This in turn weakens the exclusion in the $m_{\psi_{1,2}} - m_\phi$ plane, which we are primarily interested in. Finally, we set $|\lambda_{e1}| = |\lambda_{\mu2}| = |\lambda_{\ell\ell}|$ as required by the assumption of $\mu - e$ universality in the CMS analysis.

Note that the value of the scaling parameter $\xi$ defined in eq. (2) has no impact on the signal cross section, as the relative size of left- and right-handed couplings does not change the hierarchy between the couplings $|\lambda_{ii}|$ to different lepton flavours. This in turn means that the branching ratios of the charged mediators are independent of $\xi$. We furthermore assume the mixing between the charged mediators $\psi_1$ and $\psi_2$ to be maximal with $\theta_\psi = \pi/4$ which corresponds to the case of equal gauge eigenstate mass parameters $m_\Psi = m_\psi$.

Also note that due to the existence of two charged mediators $\psi_{1,2}$ in our model the limits from the search mentioned above cannot be straightforwardly applied to our case. As a leading order estimate we calculate the signal cross sections of the two processes $pp \to \psi_1 \bar{\psi}_1 \to \ell \bar{\ell} + \slashed{E}_T$ and $pp \to \psi_2 \bar{\psi}_2 \to \ell \bar{\ell} + \slashed{E}_T$ and compare each signal with the experimental upper limits from Reference [39] to draw exclusion contours in both the $m_{\psi_1} - m_\phi$ and the $m_{\psi_2} - m_\phi$ plane.[5] In doing so we neglect the contribution from the other mediator's pair production as well as the off-diagonal production of $\psi_1 \psi_2$ pairs, which we expect all to only marginally increase the exclusion in the above mentioned NP mass planes.

---

[5]Remember that for a maximum mixing angle $\theta_\psi = \pi/4$ the masses $m_{\psi_1}$ and $m_{\psi_2}$ are linearly connected through $m_{\psi_1} = m_{\psi_2} + \sqrt{2} y_\psi v$.

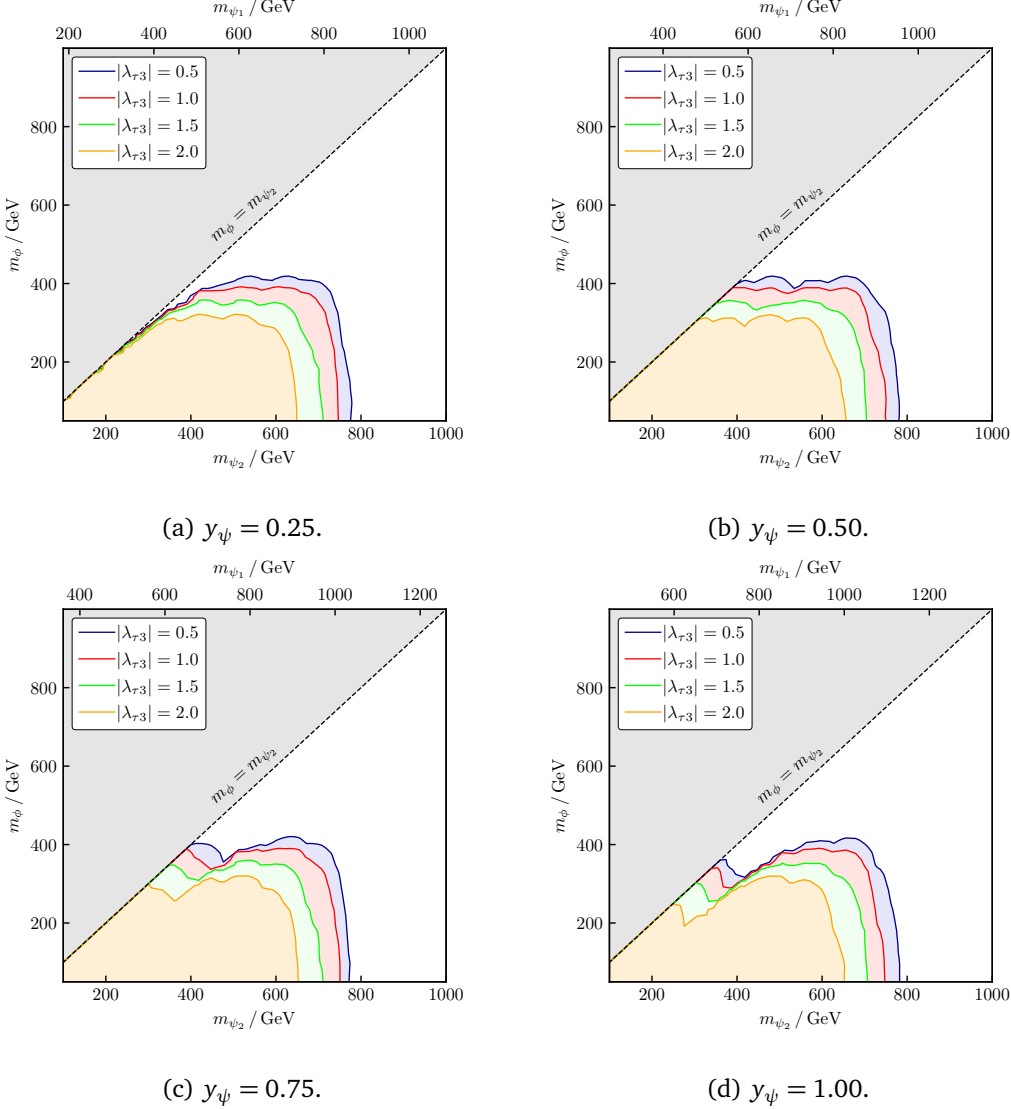

Figure 3: Constraints on the final state $\ell\bar{\ell} + \not{E}_T$ for several values of $y_\psi$, $|\lambda_{\ell\ell}| = 2.0$ and maximum mixing with $m_\Psi = m_\psi$ and $\theta_\psi = \pi/4$. The areas under the curves are excluded.

The results are shown in Figure 3. Here we overlay the exclusion in the $m_{\psi_1} - m_\phi$ as well as the $m_{\psi_2} - m_\phi$ plane in a single graph by using the linear connection between the masses of both charged mediators. In all four figures the excluded region shrinks for growing values of $|\lambda_{\tau 3}|$ as this increases the branching ratio of the charged mediators' decay into a tau-antitau pair and missing energy. The concomitant decrease of the decay rate into light lepton final states yields a smaller exclusion. While increasing the value of the Yukawa coupling $y_\psi$ has no impact on the maximum extension of the exclusion contour, it has significant implications for the exclusion of DM near the equal-mass threshold $m_\phi \approx m_{\psi_2}$. The former behaviour indicates that the contributions to the signal cross section from Higgs mediated Drell-Yan processes are negligible. The exclusion in the soft final state region is due to contributions from the heavier charged mediator $\psi_1$. Due to the mass splitting between $\psi_1$ and $\psi_2$ given as $\Delta m_\psi = m_{\psi_1} - m_{\psi_2} = \sqrt{2} y_\psi v$, the final state leptons are not produced softly if they stem from the decay of the heavier mediator $\psi_1$. In this part of the parameter space, we find that the exclusion grows for increasing Yukawa couplings up to $y_\psi \simeq 0.50$, where the strongest exclusion

is obtained, as can be seen in Figure 3b. We further see in Figure 3b, Figure 3c and Figure 3d that the exclusions in the near-degenerate region reach their maximum for $0.50 \lesssim y_\psi \lesssim 1.00$ and extend up to $m_\phi \approx m_{\psi_2} \simeq 400\,\text{GeV}$. Even larger values $y_\psi \gtrsim 1.00$ tend to reduce the exclusion in this region, since they at the same time increase the mass splitting $\Delta m_\psi$ between $\psi_1$ and $\psi_2$. This means that for such large values of $y_\psi$ even if $m_{\psi_2}$ is small the mass $m_{\psi_1}$ grows sufficiently large to suppress the $\bar\psi_1 \psi_1$ pair production cross section below the excluded range. Away from the equal mass threshold, we find that constraints from $\ell\bar\ell + \slashed{E}_T$ searches reach up to mediator masses $m_{\psi_2} \simeq 750\,\text{GeV}$, or $m_{\psi_2} \simeq 400\,\text{GeV}$ if $m_\phi \gtrsim 400\,\text{GeV}$.

## 4 Flavour physics phenomenology

Constraints from flavour physics experiments generally have a significant impact on the parameter space of flavoured DM models [20–26]. For the case of lepton-flavoured DM these constraints come from LFV decays, in particular $\ell_i \to \ell_j \gamma$, and have proven to be even stronger [21, 26] than constraints from neutral meson mixing, which are relevant for quark flavoured DM [20, 22–25]. As in our model the DM triplet couples to both right- and left-handed leptons, the NP contribution to these decays gets enhanced by contributions with a chirality flip inside the loop. In this section we carefully analyse the constraints and determine which part of our model's parameter space is consistent with experimental limits.

### 4.1 Lepton flavour violating decays

In our analysis in Reference [26] we discussed the decay rates for the LFV process shown in Figure 4 based on References [47, 48] for a generic interaction Lagrangian of the form

$$\mathcal{L}_{\text{int}} = c_{ij}^R \, \bar\ell_{Ri} \psi \phi_j + c_{ij}^L \, \bar\ell_{Li} \psi \phi_j + \text{h.c.}, \tag{14}$$

where $\psi$ is a Dirac fermion with electric charge $Q_\psi = -1$ and the fields $\phi_i$ are scalars. For the relevant branching ratios we found

$$\text{BR}(\ell_i \to \ell_j \gamma) = \frac{e^2}{64\pi} \frac{m_{\ell_i}}{\Gamma_{\ell_i}} \left( |a_{\ell_i \ell_j \gamma}^R|^2 + |a_{\ell_i \ell_j \gamma}^L|^2 \right), \tag{15}$$

with the coefficients[6]

$$a_{\ell_i \ell_j \gamma}^R = \frac{m_{\ell_i}}{16\pi^2} \sum_k \left( \frac{m_{\ell_i}}{12 m_{\phi_k}^2} c_{ik}^{R*} c_{jk}^R F(x_k) + \frac{m_\psi}{3 m_{\phi_k}^2} c_{ik}^{L*} c_{jk}^R G(x_k) \right), \tag{16}$$

$$a_{\ell_i \ell_j \gamma}^L = \frac{m_{\ell_i}}{16\pi^2} \sum_k \left( \frac{m_{\ell_i}}{12 m_{\phi_k}^2} c_{ik}^{L*} c_{jk}^L F(x_k) + \frac{m_\psi}{3 m_{\phi_k}^2} c_{ik}^{R*} c_{jk}^L G(x_k) \right), \tag{17}$$

where $x_k = m_\psi^2 / m_{\phi_k}^2$. The loop functions $F(x)$ and $G(x)$ are defined in [47, 48] and can also be found in [26]. Since our model contains two charged mediators, we obtain a total of four coefficients, which read

$$a_{\ell_i \ell_j \gamma}^{R,1} = \frac{m_{\ell_i}}{16\pi^2} \sum_k \left( \frac{m_{\ell_i} \sin^2\theta_\psi}{12 m_{\phi_k}^2} \lambda_{ik}^* \lambda_{jk} F(x_{k,1}) + \frac{m_{\psi_1} \xi^* \sin\theta_\psi \cos\theta_\psi}{3 m_{\phi_k}^2} \lambda_{ik}^* \lambda_{jk} G(x_{k,1}) \right), \tag{18}$$

$$a_{\ell_i \ell_j \gamma}^{R,2} = \frac{m_{\ell_i}}{16\pi^2} \sum_k \left( \frac{m_{\ell_i} \cos^2\theta_\psi}{12 m_{\phi_k}^2} \lambda_{ik}^* \lambda_{jk} F(x_{k,2}) - \frac{m_{\psi_2} \xi^* \sin\theta_\psi \cos\theta_\psi}{3 m_{\phi_k}^2} \lambda_{ik}^* \lambda_{jk} G(x_{k,2}) \right), \tag{19}$$

---

[6]Note that we use the convention in which the superscript refers to the chirality of the final state.

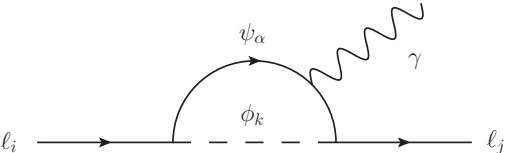

Figure 4: Feynman diagram for the LFV decay $\ell_i \to \ell_j \gamma$. The index $\alpha$ here only refers to the charged mediators, i.e. $\alpha \in \{1, 2\}$ while the indices $i, j$ and $k$ are flavour indices. The contribution from the photon coupling to one of the SM leptons is not shown.

$$a^{L,1}_{\ell_i\ell_j\gamma} = \frac{m_{\ell_i}}{16\pi^2}\sum_k\left(\frac{m_{\ell_i}|\xi|^2\cos^2\theta_\psi}{12m^2_{\phi_k}}\lambda^*_{ik}\lambda_{jk}F(x_{k,1}) + \frac{m_{\psi_1}\xi\sin\theta_\psi\cos\theta_\psi}{3m^2_{\phi_k}}\lambda^*_{ik}\lambda_{jk}G(x_{k,1})\right), \quad (20)$$

$$a^{L,2}_{\ell_i\ell_j\gamma} = \frac{m_{\ell_i}}{16\pi^2}\sum_k\left(\frac{m_{\ell_i}|\xi|^2\sin^2\theta_\psi}{12m^2_{\phi_k}}\lambda^*_{ik}\lambda_{jk}F(x_{k,2}) - \frac{m_{\psi_2}\xi\sin\theta_\psi\cos\theta_\psi}{3m^2_{\phi_k}}\lambda^*_{ik}\lambda_{jk}G(x_{k,2})\right), \quad (21)$$

where $x_{k,\alpha} = m^2_{\psi_\alpha}/m^2_{\phi_k}$. The relevant branching ratio is given in this notation as

$$\mathrm{BR}(\ell_i \to \ell_j\gamma) = \frac{e^2}{64\pi}\frac{m_{\ell_i}}{\Gamma_{\ell_i}}\left(|a^{R,1}_{\ell_i\ell_j\gamma} + a^{R,2}_{\ell_i\ell_j\gamma}|^2 + |a^{L,1}_{\ell_i\ell_j\gamma} + a^{L,2}_{\ell_i\ell_j\gamma}|^2\right). \quad (22)$$

We use these expressions to constrain the parameter space of our model in the following section.

## 4.2 Constraints from LFV decays

For the numerical analysis of constraints from LFV decays we calculate the relevant branching ratios using eq. (22) and compare them with the respective experimental bounds. The latter exist in form of 90% C.L. upper limits on the LFV branching ratios and read [49–51]

$$\mathrm{BR}(\mu \to e\gamma)_{\max} = 4.2 \times 10^{-13}, \quad (23)$$

$$\mathrm{BR}(\tau \to e\gamma)_{\max} = 3.3 \times 10^{-8}, \quad (24)$$

$$\mathrm{BR}(\tau \to \mu\gamma)_{\max} = 4.2 \times 10^{-8}. \quad (25)$$

The values for lepton masses and decay widths are taken from [52].

To obtain a rough estimate of the size of the contributions from diagrams with a chirality flip in the loop, we expand eq. (22) for $m_{\psi_{1,2}} \gg m_\phi$ while at the same time ignoring contributions from chirality preserving decays by setting the first summand in eqs. (18)–(21) to zero. We further assume maximum mixing between $\psi_1$ and $\psi_2$, i.e. we set $\theta_\psi = \pi/4$. In this limit the experimental bound on the decay $\mu \to e\gamma$ reduces to the condition

$$\sqrt{(\lambda\lambda^\dagger)_{\mu e}} \lesssim \frac{1}{2200\,\mathrm{TeV}}\sqrt{\frac{m_{\psi_1}m_{\psi_2}}{y_\psi|\xi|}}, \quad (26)$$

which for a NP scale $m_{\psi_1}$ of order $\mathcal{O}(1\,\mathrm{TeV})$ together with an order $\mathcal{O}(1)$ Yukawa coupling $y_\psi$ yields an upper limit on the couplings of

$$\sqrt{(\lambda\lambda^\dagger)_{\mu e}} \lesssim 3.7 \times \frac{10^{-4}}{\sqrt{|\xi|}}. \quad (27)$$

While this estimate gives us a decent understanding of the strength of the LFV constraint, in our subsequent numerical analysis we use the full quantitative expression of Section 4.1.

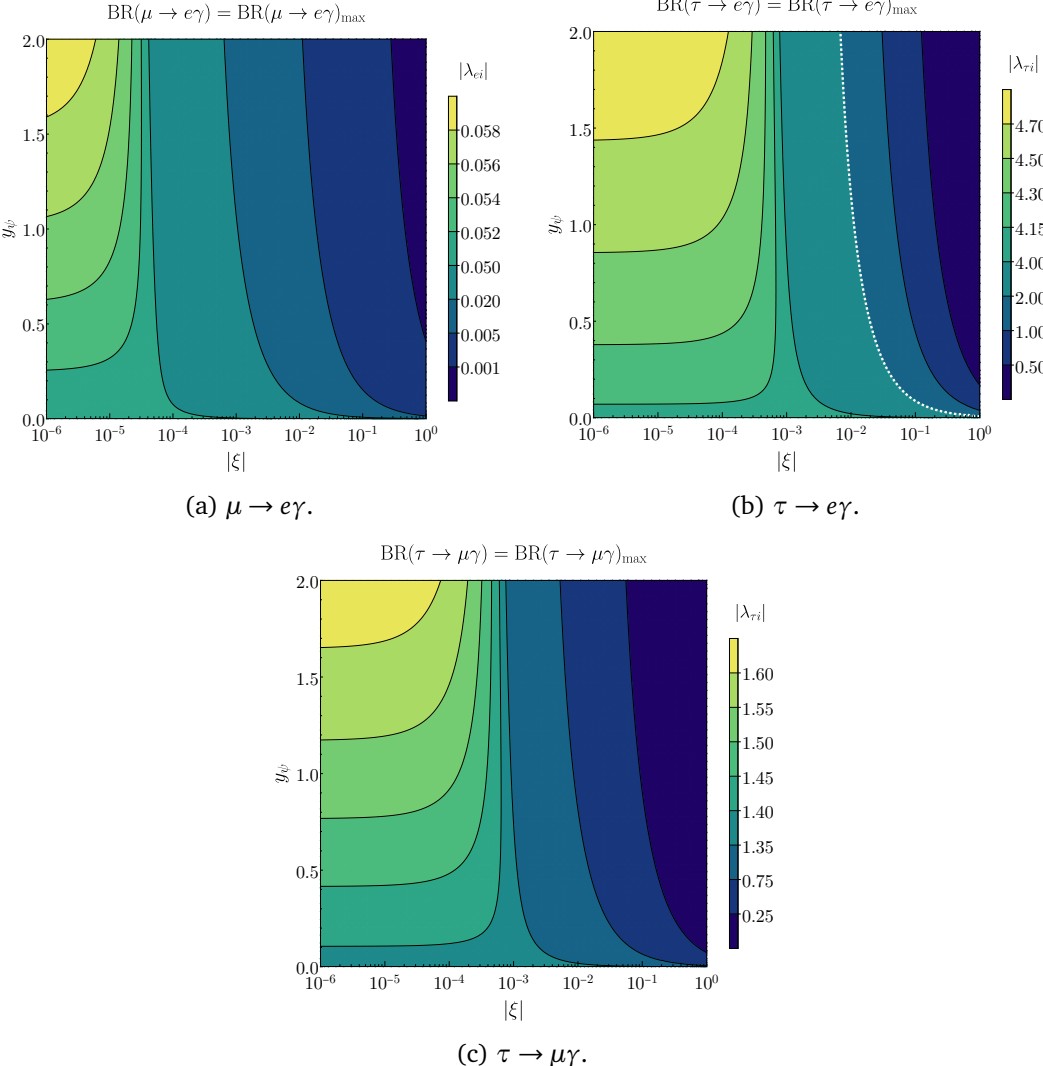

Figure 5: Constraints from LFV decays on the coupling matrix $\lambda$. In all three plots maximum mixing with $\theta_\psi = \pi/4$ is assumed. We further set $m_\phi = 200\,\text{GeV}$, $m_{\psi_2} = 1300\,\text{GeV}$ and the value of $m_{\psi_1}$ varies according to the value of $y_\psi$.

To get a more thorough insight on how strongly the LFV decays constrain the coupling matrix $\lambda$, in the contour plots of Figure 5 we show the maximum possible couplings $|\lambda_{\ell i}|$ for varying values of $y_\psi$ and $|\xi|$ by comparing the full expression from eq. (22) with the respective experimental upper limit quoted above. To this end we assume universal couplings $|\lambda_{\ell i}|$ of all DM flavours $i = 1, 2, 3$. Concerning the mass spectrum, we take the mixing to be maximal, $\theta_\psi = \pi/4$, and set $m_{\psi_2} = 1300\,\text{GeV}$ as well as $m_\phi = 200\,\text{GeV}$ in all three figures. Depending on the value of $y_\psi$, this leads to a maximal mass of the heavier mediator $\psi_1$ of roughly $m_{\psi_1} = 2000\,\text{GeV}$.

In Figure 5a we have set the DM–muon couplings to $|\lambda_{\mu i}| = 1$ to not suppress NP effects in $(g-2)_\mu$. Thus, Figure 5a shows the largest possible values for the DM–electron couplings $|\lambda_{ei}|$ which can be reconciled with the experimental upper bound on the strongly constrained LFV decay $\mu \to e\gamma$. As expected and as our rough estimate from eq. (26) already suggests, the upper limit on $|\lambda_{ei}|$ carries a strong dependence on $|\xi|$ while the $y_\psi$ dependence is rather mild for values of $|\xi| \sim \mathcal{O}(10^{-4} - 10^0)$ in which the chirality flipped contributions dominate. This is due to our fixing of the light charged mediator mass to $m_{\psi_2} = 1300\,\text{GeV}$. Growing

values of $y_\psi$ increase the mass splitting $\Delta m_\psi = m_{\psi_1} - m_{\psi_2}$ since they increase the value of $m_{\psi_1}$. As the branching ratio from eq. (22) is roughly proportional to this mass splitting, this leads to a growth of the branching ratio, while the growing value of $m_{\psi_1}$ at the same time suppresses the contributions coming from diagrams with $\psi_1$ in the loop. In combination we find that growing values of $y_\psi$ still lead to a mild overall growth of BR($\mu \to e\gamma$). For values $|\xi| \lesssim 0.5 \times 10^{-4}$ the right-handed chirality-preserving contributions, i.e. the first summands of eq. (18) and eq. (19) are dominant, as all other contributions are suppressed by the small value of $|\xi|$. In this region increasing values of $y_\psi$ allow for larger couplings $|\lambda_{ei}|$ as $m_{\psi_1}$ grows with $y_\psi$, which in turn suppresses the chirality preserving contribution $a_{\mu e\gamma}^{R,1}$ through the loop function $F$. Figure 5a also shows that depending on the choice of $y_\psi$ and $|\xi|$ the DM–electron couplings vary between values $|\lambda_{ei}| \sim \mathcal{O}(10^{-4} - 10^{-1})$. As smaller mediator masses demand even smaller values of $|\lambda_{ei}|$, we will restrict the range of these couplings to $|\lambda_{ei}| \in [10^{-6}, 10^{-1}]$ when scanning over the parameter space of our model in the remainder of our analysis.

Figure 5b displays the constraints that the LFV decay $\tau \to e\gamma$ places on the parameter space of our model. Here we set the DM–electron couplings to $|\lambda_{ei}| = 0.1$ in order to quantify how strongly this decay constrains the DM–tau couplings. The $y_\psi$ and $|\xi|$ dependence are qualitatively the same as in Figure 5a with the only difference that the chirality-preserving contribution starts to dominate for values $|\xi| \lesssim 10^{-3}$ in this case. This is due to the fact that the latter is proportional to $m_{\ell_i}^2$ while the muon mass is roughly a factor of 17 smaller than the tau mass. The white dashed line in Figure 5b indicates in which part of the $|\xi| - y_\psi$ plane we expect constraints on $|\lambda_{\tau i}|$ from the decay $\tau \to e\gamma$, as we have generally limited the couplings to $|\lambda_{ij}| \in [0, 2]$ in Section 2. We find that this LFV decay only constrains the DM–tau couplings for values $|\xi| \gtrsim 5 \times 10^{-2}$.

The constraints from the LFV decay $\tau \to \mu\gamma$ are shown in Figure 5c. In order to not suppress NP effects in $(g-2)_\mu$ we have once more set $|\lambda_{\mu i}| = 1$. Again the $y_\psi$ and $|\xi|$ dependence is the same as for the previous cases. We find that in spite of its comparably weak upper limit this decay restricts the DM–tau couplings to the range $|\lambda_{\tau i}| \sim \mathcal{O}(10^{-1} - 10^0)$.

## 5 Precision measurements

An important feature of lepton-flavoured DM models is that they are subject to limits from precision measurements of leptonic electric dipole moments (EDM) $d_\ell$ and anomalous magnetic dipole moments (MDM) $a_\ell$. We found in [26] that for purely right-handed interactions between the SM and DM these constraints are not relevant for masses allowed by collider searches due to the lack of a chirality flip enhancement. In contrast, in the present study the DM triplet is coupled to both right- and left-handed leptons, so that NP contributions to $d_\ell$ and $a_\ell$ can become sizeable even for large mediator masses allowed by collider searches. While this opens the possibility to explain the discrepancy between theory and experiment in $a_\mu$, it also constrains the parameter space of the model through the EDM and MDM of the electron. We use this section to discuss the latter, while NP effects in $a_\mu$ will be treated separately in Section 9. We start with a general discussion of the possible NP contributions to both $d_\ell$ and $a_\ell$ and then present a numerical analysis specific to our model.

### 5.1 Lepton EDM and MDM

The Feynman diagram inducing one-loop contributions to the EDM $d_\ell$ and MDM $a_\ell$ is obtained when setting $i = j$ in the radiative process $\ell_i \to \ell_j\gamma$ illustrated in Figure 4. Following our

notation from the previous section and [53] we can write for its amplitude

$$\mathcal{M}_{\ell_i \ell_i \gamma} = \frac{e}{2m_{\ell_i}} \epsilon^{*\alpha} \bar{u}_{\ell_i} \left[ i\sigma_{\beta\alpha} q^\beta \left( a^R_{\ell_i \ell_i \gamma} P_L + a^L_{\ell_i \ell_i \gamma} P_R \right) \right] u_{\ell_i} + \epsilon^{\mu*} \bar{u}_{\ell_i} \left[ \sigma_{\nu\mu} \gamma_5 q^\nu d_{\ell_i} \right] u_{\ell_i}, \qquad (28)$$

where $\sigma_{\beta\alpha} = i[\gamma_\alpha, \gamma_\beta]/2$, $\epsilon$ is the photon polarization vector, $q$ is the photon momentum and $P_{R/L} = (1 \pm \gamma_5)/2$ are projection operators. For the generic Lagrangian introduced in eq. (14) the NP contribution $\Delta a_{\ell_i}$ to the MDM $a_{\ell_i}$ and the EDM $d_{\ell_i}$[7] of the lepton $\ell_i$ then read [53,56]

$$\begin{aligned}
\Delta a_{\ell_i} &= a^R_{\ell_i \ell_i \gamma} + a^L_{\ell_i \ell_i \gamma} \\
&= \frac{m_{\ell_i}}{16\pi^2} \sum_k \left( \frac{m_{\ell_i}}{12 m^2_{\phi_k}} (|c^R_{ik}|^2 + |c^L_{ik}|^2) F(x_k) + \frac{2m_\psi}{3m^2_{\phi_k}} \mathrm{Re}\left[ c^{L*}_{ik} c^R_{ik} \right] G(x_k) \right),
\end{aligned} \qquad (29)$$

and

$$d_{\ell_i} = -\frac{e}{16\pi^2} \sum_k \frac{m_\psi}{3m^2_{\phi_k}} \mathrm{Im}\left[ c^R_{ik} c^{L*}_{ik} \right] G(x_k). \qquad (30)$$

Here, the coefficients $a^{R/L}_{\ell_i \ell_i \gamma}$ as well as the loop functions $F$ and $G$ are the same as in eq. (16) and eq. (17). Note that an EDM $d_{\ell_i}$ is only induced if the couplings defined in eq. (14) satisfy

$$\mathrm{Im}\left[ c^R c^{L*} \right] \neq 0, \qquad (31)$$

i.e. if the Lagrangian $\mathcal{L}_{\mathrm{int}}$ from eq. (14) violates CP symmetry [57].

As there are two charged mediators $\psi_1$ and $\psi_2$ in our model, we define the two coefficients

$$\Delta a^1_{\ell_i} = a^{R,1}_{\ell_i \ell_i \gamma} + a^{L,1}_{\ell_i \ell_i \gamma}, \qquad (32)$$

$$\Delta a^2_{\ell_i} = a^{R,2}_{\ell_i \ell_i \gamma} + a^{L,2}_{\ell_i \ell_i \gamma}, \qquad (33)$$

which when mapping the expressions from above to our model read

$$\Delta a^1_{\ell_i} = \frac{m_{\ell_i}}{16\pi^2} \sum_k \left( \frac{m_{\ell_i} |\lambda_{ik}|^2}{12 m^2_{\phi_k}} (s^2_\theta + |\xi|^2 c^2_\theta) F(x_{k,1}) + \frac{2m_{\psi_1} c_\theta s_\theta |\lambda_{ik}|^2}{3m^2_{\phi_k}} \mathrm{Re}\, \xi\, G(x_{k,1}) \right), \qquad (34)$$

$$\Delta a^2_{\ell_i} = \frac{m_{\ell_i}}{16\pi^2} \sum_k \left( \frac{m_{\ell_i} |\lambda_{ik}|^2}{12 m^2_{\phi_k}} (c^2_\theta + |\xi|^2 s^2_\theta) F(x_{k,2}) - \frac{2m_{\psi_2} c_\theta s_\theta |\lambda_{ik}|^2}{3m^2_{\phi_k}} \mathrm{Re}\, \xi\, G(x_{k,2}) \right). \qquad (35)$$

Here we have used $s_\theta = \sin\theta_\psi$ and $c_\theta = \cos\theta_\psi$ for brevity of notation. The total NP contribution to $a_{\ell_i}$ is then defined as

$$\Delta a_{\ell_i} = \Delta a^1_{\ell_i} + \Delta a^2_{\ell_i}. \qquad (36)$$

Similarly, in our model the EDMs $d_{\ell_i}$ can be defined as $d_{\ell_i} = d^1_{\ell_i} + d^2_{\ell_i}$, with

$$d_{\ell_i} = -\frac{e}{16\pi^2} \sum_k \frac{c_\theta s_\theta |\lambda_{ik}|^2}{3m^2_{\phi_k}} \mathrm{Im}\, \xi \left( m_{\psi_2} G(x_{k,2}) - m_{\psi_1} G(x_{k,1}) \right). \qquad (37)$$

Note that a negative scaling parameter $\xi$ implies positive contributions to both the lepton MDMs and EDMs. As we are ultimately interested in solving the $(g-2)_\mu$ anomaly which requires sizeable positive NP contributions to $a_\mu$, we only consider the case $\xi < 0$ throughout the rest of this analysis. In the following we use the expressions provided above to determine the constraints that precision measurements of the electron EDM and MDM place on our model's parameter space.

---

[7]Note that since leptonic EDMs arise at the four-loop level [54] in the SM, estimates [55] provide an upper limit of $d^{\mathrm{SM}}_e < 10^{-38} e\,\mathrm{cm}$. We hence ignore SM contributions to the lepton EDMs $d_{\ell_i}$.

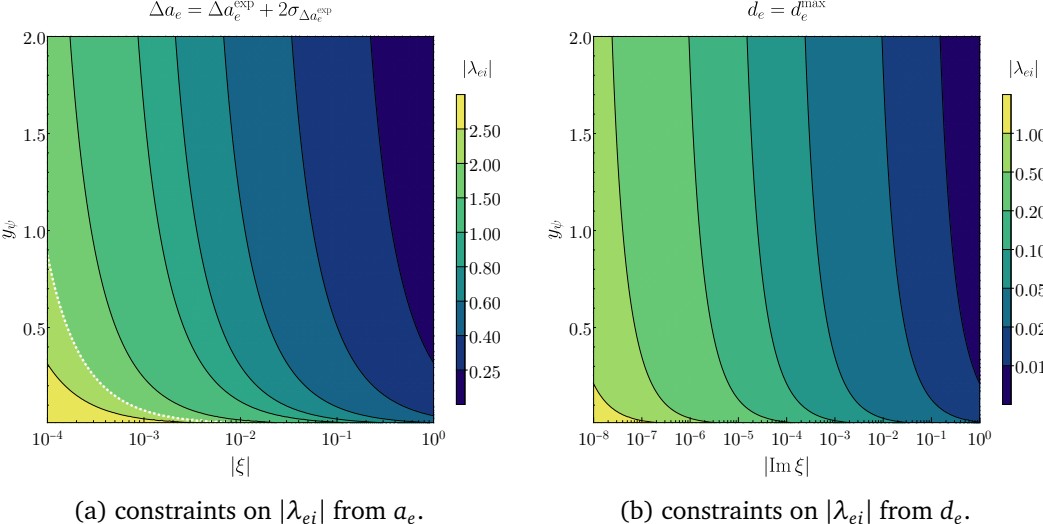

(a) constraints on $|\lambda_{ei}|$ from $a_e$.      (b) constraints on $|\lambda_{ei}|$ from $d_e$.

Figure 6: Constraints from precision measurements on the coupling matrix $\lambda$. In both panels maximum mixing with $\theta_\psi = \pi/4$ is assumed. We further set $m_\phi = 200\,\text{GeV}$, $m_{\psi_2} = 1300\,\text{GeV}$ and the value of $m_{\psi_1}$ varies according to the value of $y_\psi$. The white dashed line in the left figure shows the contour with $|\lambda_{ei}| = 2.0$.

## 5.2 Constraints from dipole moments

The most stringent constraints from precision tests of the SM exist for the electron EDM $d_e$ and MDM $a_e$. Constraints on the muon EDM $d_\mu$ [58] and tau EDM $d_\tau$ [59] are ten orders of magnitude weaker than the current 90% C.L. upper limit on the electron EDM $d_e$, which reads [60]

$$d_e^{\max} = 1.1 \times 10^{-29} e\,\text{cm}. \tag{38}$$

Concerning MDMs, the tau MDM $a_\tau$ has not been measured precisely enough yet [52,61,62] to provide a meaningful constraint on NP contributions. In spite of having been measured at a very high precision [63], the electron anomalous magnetic moment $a_e$ on the other hand is subject to a tension caused by disagreeing measurements of the fine-structure constant $\alpha_{\text{em}}$. Predicting $a_e^{\text{SM}}$ based on a measurement [64] of $\alpha_{\text{em}}$ using $^{133}$Cs atoms yields a difference of [65]

$$\Delta a_e^{\text{exp}}(\text{Cs}) = (-8.8 \pm 3.6) \times 10^{-13}, \tag{39}$$

which corresponds to a deviation of $2.4\sigma$ between theory and experiment. However, predicting $a_e^{\text{SM}}$ based on a measurement [66] of $\alpha_{\text{em}}$ in $^{87}$Rb atoms yields [67]

$$\Delta a_e^{\text{exp}}(\text{Rb}) = (4.8 \pm 3.0) \times 10^{-13}, \tag{40}$$

corresponding to a deviation of $1.6\sigma$ in the opposite direction. Due to our choice $\xi < 0$, the NP contributions to $a_e$ are positive in our model. Hence, as a conservative approach we use the limit from eq. (40) based on the measurement of $\alpha_{\text{em}}$ using $^{87}$Rb atoms in order to constrain the DM–electron couplings further.

For the numerical analysis of constraints from electron dipole moments we use the same approach as for the flavour constraints in Section 4. To study the bounds that the electron MDM $a_e$ and EDM $d_e$ place on the DM–electron couplings, we have set them to a universal value $|\lambda_{ei}|$ and generated the contour plots shown in Figure 6. In both figures we have again set $m_\phi = 200\,\text{GeV}$, $m_{\psi_2} = 1300\,\text{GeV}$ and use maximum mixing with $\theta_\psi = \pi/4$. The mass $m_{\psi_1}$ varies according to the value of $y_\psi$.

In Figure 6a the contours show which values of $|\lambda_{ei}|$ are maximally allowed to stay in the $2\sigma$-band of $\Delta a_e^{\text{exp}}$ in the $|\xi|-y_\psi$ plane. The white dashed line again shows the contour with the maximally allowed value of 2.0 for $|\lambda_{ei}|$. We find that in comparison to the LFV constraints, restrictions on $|\lambda_{ei}|$ from measurements of the electron anomalous magnetic moment are less severe. Satisfying the constraints from LFV decays while allowing for order $\mathcal{O}(1)$ DM–muon couplings already forces the DM–electron couplings to be so small that restrictions on $|\lambda_{ei}|$ from $a_e$ are rendered irrelevant.

The constraints from measurements of the electron EDM $d_e$ are shown in Figure 6b. Here, the contours show the maximum possible values for $|\lambda_{ei}|$ to still reconcile with the experimental upper limit $d^{\text{exp}}$ in the $|\text{Im}\xi|-y_\psi$ plane. Since a non-zero EDM requires the violation of CP symmetry, we here show the absolute value of the imaginary part of the scaling parameter $\xi$ instead of the absolute value of $\xi$ itself. We find that a relative phase between the DM triplet's coupling to right-handed and left-handed leptons, which is generated by $\text{Im}\xi$ is strongly constrained by the electron EDM $d_e$. For values $|\lambda_{ei}| \sim \mathcal{O}(10^{-4}-10^{-1})$ which are necessary to fulfil the flavour constraints it allows for $|\text{Im}\xi| \sim \mathcal{O}(10^{-4}-10^{0})$. We conclude that in spite of the strength of the EDM constraint, it still allows for a relative CP phase between the left-handed and right-handed couplings to leptons provided that the absolute strength of the coupling to electrons, $|\lambda_{ei}|$, is sufficiently suppressed.

## 5.3 Electroweak precision observables

Before concluding this section we want to also comment on possible constraints on the parameter space of our model coming from electroweak precision observables.

Firstly, the new particles contribute to the oblique parameters $S$ and $T$, with the largest contribution from the electroweak doublet $\Psi$ [68,69]. However, as our model does not violate custodial symmetry, the latter are small. The contributions to $S$ are moderate as well, since the electroweak interactions of $\Psi$ are vectorlike. In conjunction with the NP scale well above the electroweak scale, as found in our global analysis (see Section 8), we estimate the currently available constraints to not be competitive.

Secondly, the new interactions with the Higgs may lead to virtual corrections to Higgs boson couplings. Like throughout the rest of our analysis, we neglect the impact of the Higgs portal coupling $\lambda_{H\phi}$ here and focus on the new Yukawa coupling $y_\psi$. In Reference [68], a very similar model setup with a fermionic doublet and singlet was studied. In that analysis the respective Yukawa coupling $y$ was found to be unconstrained by current data in the region $y \leq 3$. With our choice of parameter range $y_\psi \leq 2$, see Equation (4), our model is thus safe from current Higgs data.

Finally, the leptonic NP interactions of our model can induce vertex corrections to the couplings of leptons to electroweak gauge bosons at the one-loop level. These corrections in turn have an impact on the Fermi constant $G_F$ as well as the $Z$ boson couplings to leptons which potentially poses a problem for the global electroweak fit. However, we estimate these contributions to be safely small since they are suppressed by a loop factor as well as the NP scale $m_{\text{NP}} \sim \mathcal{O}(1\,\text{TeV})$.

We note that due to the significant improvements expected in Higgs and electroweak precision data from future hadron and lepton colliders, these observables might become a powerful tool to test our model. A detailed study of the reach of future colliders is beyond the scope of our work and we refer the reader to [68,69] for results within similar models.

# 6 DM relic density

As we are not only proposing the model presented in Section 2 as a solution to the $(g-2)_\mu$ anomaly but also as a viable DM model, its parameter space is also subject to constraints from cosmological determinations of the DM relic density [70]

$$\Omega_c h^2 = 0.120 \pm 0.001 \,. \tag{41}$$

In this section we discuss the impact of the required DM relic density on the parameter space of our model.

## 6.1 DM thermal freeze-out

For the analysis of the relic density constraints we assume a thermal freeze-out of DM at $T_f \approx m_{\phi_3}/20$. At this time in the early universe the DM production and annihilation rates approach zero, leading to a decoupling of the dark species from thermal equilibrium. Hence, the co-moving number density of DM resulting from this freeze-out process depends on the effective annihilation rate of DM $\langle \sigma v \rangle_{\text{eff}}$.

As the splittings between the masses[8] $m_{\phi_i}$ determine the relative number density of the different dark flavours $\phi_i$ at $T_f$, their contributions to the freeze-out also depends on the mentioned splittings. The most generic dynamics of flavoured DM freeze-out is rather involved and is the subject of a separate ongoing study [71]. In this work we are interested in the main phenomenological features of the model which can be captured by the study of two simplifying benchmark scenarios, following the approach in [20–26]. This also allows for a direct comparison of our results with the ones for lepton-flavoured scalar DM coupling only to right-handed leptons [26]. We hence restrict our study to the following two benchmark scenarios for the freeze-out.

- We call the scenario with a near-degenerate mass spectrum $m_{\phi_i}$ the *Quasi-Degenerate Freeze-Out* (QDF). As the splittings between the two heavier states and the lightest state are assumed to be very small in this scenario, the co-moving number densities of all three dark flavours are roughly equal at the time $T_f$ such that all of them equally contribute to the freeze-out. As the mass splittings between the heavy states $\phi_{1,2}$ and the lightest state $\phi_3$ are not zero, the heavy states eventually decay into the lightest state at lower temperatures after the freeze-out. Note that these decays still happen at a sufficient rate to not affect big bang nucleosynthesis or yield energy injections into the cosmic microwave background [20].

- We further consider a benchmark scenario in which the masses of the lightest and the heavier states are significantly split. In this scenario, which we refer to as the *Single-Flavour Freeze-Out* (SFF) scenario, the lifetime of the heavy states $\phi_1$ and $\phi_2$ is short in comparison to the time of the freeze-out. As the rate of flavour changing scattering processes are much larger than the Hubble rate,[9] a relative equilibrium between different dark species is maintained, but the number density of the heavy states is strongly suppressed by a Boltzmann factor with the respective mass splitting as its argument. Thus, only the lightest state $\phi_3$ contributes to the freeze-out in this scenario.

---

[8]In order to avoid ambiguities, the mass parameterss $m_{\phi_i}$ include potential loop corrections, i. e. they are renormalised on-shell masses.

[9]We have checked the accuracy of this approximation in an ongoing analysis [71] by numerically solving the coupled three-flavour Boltzmann equations.

Numerically we define the two scenarios through the mass splittings

$$\Delta m_{i3} = \frac{m_{\phi_i}}{m_{\phi_3}} - 1 \,, \tag{42}$$

between the heavier states with $i \in \{1, 2\}$ and the lightest state $\phi_3$. In the QDF scenario $\Delta m_{i3}$ may not be larger than 1%, while we demand $10\% < \Delta m_{i3} < 30\%$ for the SFF scenario, where the upper limit is applied in order to keep our results comparable to previous studies in the DMFV framework, see Section 2.2. Remember that as discussed in Section 2.2 in contrast to the studies performed in [20–26] the splittings $\Delta m_{i3}$ are not parametrized through the DMFV spurion expansion in our model, due to the absence of a flavour symmetry. Thus, they are independent of the coupling $\lambda$.

In Figure 7 we gather possible tree-level annihilations of the NP fields into SM fields. Only if the virtual particle in Figure 7a is $\psi_0$, the DM pair $\phi_i \phi_j$ annihilates into a pair of neutrinos $\bar\nu_k \nu_l$. In the coannihilation diagram of Figure 7b a neutrino in the final state is only produced for $\beta = 0$ together with a $W$ boson for $\alpha \in \{1, 2\}$ or a $Z$ for $\alpha = 0$. The cases $\alpha = 1, \beta = 2$ and vice versa produce either of the final states $\ell_j Z$ or $\ell_j h$, while the case $\alpha = \beta \in \{1, 2\}$ can additionally produce the final state $\ell_j \gamma$. The annihilation between $\psi_\alpha$ and $\psi_\beta$ can produce final states with either two charged leptons, two neutrinos or one neutrino and one charged lepton.

Note that the coannihilation channels shown in Figure 7b suffer from a Boltzmann suppression by the factor

$$k_\alpha = e^{-\frac{m_{\psi_\alpha} - m_{\phi_3}}{T_f}} \simeq e^{-20\frac{m_{\psi_\alpha} - m_{\phi_3}}{m_{\phi_3}}} \,, \tag{43}$$

while the annihilations of Figure 7c receive an even stronger suppression by $k_\alpha k_\beta$. These processes are thus irrelevant outside of the highly fine-tuned parameter region $m_{\phi_3} \simeq m_{\psi_2}$ which we omit in our analysis. Also note that there exist additional annihilation processes that we do not show in Figure 7, related to the Higgs portal coupling $\lambda_{H\phi}$. The annihilation of DM into a pair of Higgs bosons is governed by this quartic coupling and is proportional to $\lambda_{H\phi}^2$. It also gives rise to the annihilation of a pair of dark scalars into a virtual Higgs boson in the $s$-channel which subsequently decays into SM fermions. Annihilations into a top–antitop pair in this channel are thus proportional to $\lambda_{H\phi}^2 y_t^2$ and can generally become sizable, due to the large top Yukawa coupling. We follow our arguments from [26] and assume the (renormalised) coupling $\lambda_{H\phi}$ to be sufficiently small such that these diagrams can be neglected. Recall that in our analysis we are primarily interested in the structure of the flavour-violating coupling matrix $\lambda$.

We are thus left with the $t$-channel annihilation processes shown in Figure 7a. Its total flavour-averaged squared amplitude reads

$$\overline{|M|^2} = \overline{|M_0|^2} + \overline{|M_1|^2} + \overline{|M_2|^2} + 2\,\mathrm{Re}\left(\overline{M_{12}}\right) \,, \tag{44}$$

where the index corresponds to the index of the mediator $\psi_\alpha$ exchanged in the $t$-channel. The expressions for the individual contributions $M_\alpha$ and the interference term $\overline{M_{12}}$ are given as

$$\overline{|M_0|^2} = \frac{1}{9} \sum_{ij} \sum_{kl} \frac{|\lambda_{ik}|^2 |\lambda_{jl}|^2}{(t - m_{\psi_0}^2)^2} f_{ij}^0 \,, \tag{45}$$

$$\overline{|M_1|^2} = \frac{1}{9} \sum_{ij} \sum_{kl} \frac{|\lambda_{ik}|^2 |\lambda_{jl}|^2}{(t - m_{\psi_1}^2)^2} f_{ijkl}^1 \,, \tag{46}$$

$$\overline{|M_2|^2} = \frac{1}{9} \sum_{ij} \sum_{kl} \frac{|\lambda_{ik}|^2 |\lambda_{jl}|^2}{(t - m_{\psi_2}^2)^2} f_{ijkl}^2 \,, \tag{47}$$

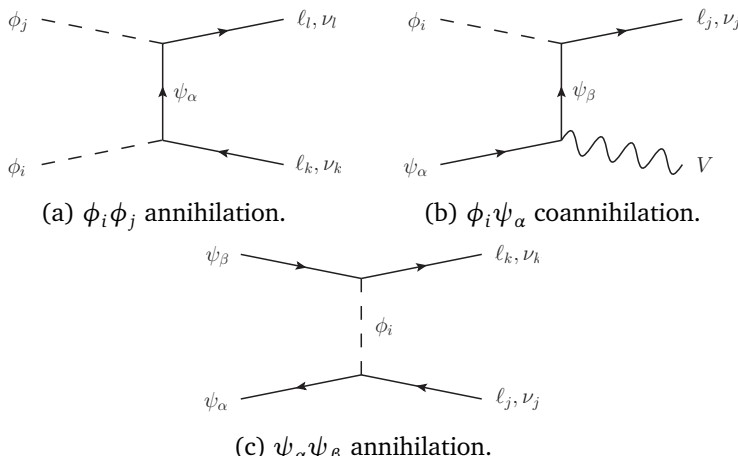

(a) $\phi_i \phi_j$ annihilation.   (b) $\phi_i \psi_\alpha$ coannihilation.

(c) $\psi_\alpha \psi_\beta$ annihilation.

Figure 7: Representative Feynman diagrams for annihilations of the new particles into SM matter. Here, $V$ represents any of the SM electroweak bosons $\gamma, W, Z$ and $h$ (we use a wiggly line as most of them are vector bosons).

$$\overline{M_{12}} = \frac{1}{9} \sum_{ij} \sum_{kl} \frac{|\lambda_{ik}|^2 |\lambda_{jl}|^2}{(t - m_{\psi_1}^2)(t - m_{\psi_2}^2)} f_{ijkl}^{12}, \tag{48}$$

with the functions $f^\alpha$ defined as

$$f_{ij}^0 = |\xi|^4 \left( \left( m_{\phi_j}^2 - t \right) \left( t - m_{\phi_i}^2 \right) - ts \right), \tag{49}$$

$$f_{ijkl}^1 = A_{ijkl} \left( s_\theta^4 + |\xi|^4 c_\theta^4 \right) + c_\theta^2 s_\theta^2 \left( 2|\xi|^2 C_{kl} + m_{\psi_1}^2 D_{kl} \right)$$
$$+ 2 c_\theta s_\theta m_{\psi_1} \mathrm{Re} B_{ijkl} \left( s_\theta^2 + |\xi|^2 c_\theta^2 \right), \tag{50}$$

$$f_{ijkl}^2 = A_{ijkl} \left( c_\theta^4 + |\xi|^4 s_\theta^4 \right) + c_\theta^2 s_\theta^2 \left( 2|\xi|^2 C_{kl} + m_{\psi_2}^2 D_{kl} \right)$$
$$- 2 c_\theta s_\theta m_{\psi_2} \mathrm{Re} B_{ijkl} \left( c_\theta^2 + |\xi|^2 s_\theta^2 \right), \tag{51}$$

$$f_{ijkl}^{12} = s_\theta^2 c_\theta^2 \left( A_{ijkl} (1 + |\xi|^4) - m_{\psi_1} m_{\psi_2} D_{kl} \right) + C_{kl} |\xi|^2 \left( c_\theta^4 + s_\theta^4 \right)$$
$$+ c_\theta s_\theta \left( B_{ijkl} c_\theta^2 \left( m_{\psi_1} - |\xi|^2 m_{\psi_2} \right) - B_{ijkl}^* s_\theta^2 \left( m_{\psi_2} - |\xi|^2 m_{\psi_1} \right) \right). \tag{52}$$

Here we have again used $s_\theta = \sin \theta_\psi$ and $c_\theta = \cos \theta_\psi$ for brevity of notation, and the indices $i, j, k$ and $l$ are flavour indices. The functions $A_{ijkl}, B_{ijkl}, C_{kl}$ and $D_{kl}$ depend on the masses $m_{\phi_i}, m_{\phi_j}, m_{\ell_k}$ and $m_{\ell_l}$ as well as the Mandelstam variables $s = (p_1 + p_2)^2$ and $t = (p_1 - p_3)^2$. Their full expressions can be found in Appendix A.

In order to constrain our model based on the observed DM relic density, we compute the low-velocity expansion [72, 73] of the effective thermally averaged annihilation cross section

$$\langle \sigma v \rangle_{\mathrm{eff}} = \frac{1}{2} \langle \sigma v \rangle = \frac{1}{2} \left( a + b \langle v^2 \rangle + \mathcal{O}(\langle v^4 \rangle) \right), \tag{53}$$

with $\langle v^2 \rangle = 6 T_f / m_{\phi_3} \simeq 0.3$. The factor of two for the conversion between $\langle \sigma v \rangle_{\mathrm{eff}}$ and $\langle \sigma v \rangle$ is due to $\phi$ being a complex scalar. The coefficients $a$ and $b$ are calculated for equal initial and distinct final state masses using the techniques provided in [73, 74]. Note that using equal initial state masses $m_{\phi_i} = m_{\phi_j}$ is justified in both freeze-out scenarios we consider, as the QDF scenario is defined through near-degenerate masses $m_{\phi_i} \approx m_{\phi_j}$ of different dark flavours, and since the only flavour contributing to the freeze-out in the SFF scenario is $\phi_3$. Thus, in the latter case, the mass parameters $m_{\phi_i}$ and $m_{\phi_j}$ in the functions $A_{ijkl}$ and $B_{ijkl}$ need to be replaced with $m_{\phi_3}$ and the sum over initial state flavours as well as the averaging factor of 1/9

need to be omitted. In what follows we therefore use $m_{\phi_3}$ whenever we refer to the DM mass in both freeze-out scenarios. In spite of the fact that setting the masses of charged leptons to zero is a very good approximation, we use the expressions for $a$ and $b$ with the full final state mass dependence in our numerical analysis.

In contrast to our findings in [26], the annihilation rate is no longer $p$-wave suppressed in this model for equal initial and zero final state masses. The mentioned $p$-wave suppression in the model of [26] is due to a chirality suppression [75], and thus adding couplings to left-handed leptons trivially lifts this suppression [76]. The leading contribution is then given by the $s$-wave term and reads

$$a = \frac{1}{9} \sum_{ij} \sum_{kl} \frac{|\lambda_{ik}|^2 |\lambda_{jl}|^2}{16\pi m_{\phi_3}^2} \frac{(\mu_2 - \mu_1)^2 (\mu_1 \mu_2 - 1)^2 |\xi|^2 \sin^2 2\theta_\psi}{\left(1 + \mu_1^2\right)^2 \left(1 + \mu_2^2\right)^2}, \tag{54}$$

which due to eq. (2) depends on the scaling parameter $\xi$. Here we have used $\mu_\alpha = m_{\psi_\alpha}/m_{\phi_3}$. The $p$-wave contribution for a non-vanishing $\xi$ can be found in Appendix A. If the left-handed coupling is suppressed, i.e. if $\xi$ approaches zero, we re-encounter the aforementioned $p$-wave suppression of the annihilation rate. In this case the coefficients read

$$a = 0, \tag{55}$$

$$b = \frac{1}{9} \sum_{ij} \sum_{kl} \frac{|\lambda_{ik}|^2 |\lambda_{jl}|^2}{32\pi m_{\phi_3}^2} \frac{\left(2 + \mu_1^2 + \mu_2^2 + \left(\mu_1^2 - \mu_2^2\right) \cos 2\theta_\psi\right)^2}{\left(1 + \mu_1^2\right)^2 \left(1 + \mu_2^2\right)^2}, \tag{56}$$

which in the limit of equal charged mediator masses $m_{\psi_1} = m_{\psi_2}$, i.e. in the limit $y_\psi = 0$, reduces to the expressions found in our analysis in [26].[10]

## 6.2 Constraints from the DM relic density

In order to determine the bounds from the observed DM relic density on the DM–lepton coupling $\lambda$, we calculate the effective thermally averaged annihilation cross section through the partial wave expansion in eq. (53) and compare with the experimental limit on $\langle \sigma v \rangle_{\text{eff}}$. The latter is derived based on the DM relic density from eq. (41). It is roughly constant for DM masses $m_{\phi_3} > 10\,\text{GeV}$ and reads [77, 78]

$$\langle \sigma v \rangle_{\text{eff}}^{\text{exp}} = 2.2 \times 10^{-26}\,\text{cm}^3\,\text{s}^{-1}. \tag{57}$$

In the numerical analysis we calculate the annihilation rate for random points of the parameter space. When generating random points we restrict the DM–electron couplings to $|\lambda_{ei}| \in [10^{-6}, 10^{-1}]$ in order to comply with the flavour constraints without precluding a solution to the $(g-2)_\mu$ anomaly. We further demand that the annihilation rate equals the experimental value from above within a 10% tolerance region. The lepton masses are again adopted from [52]. In terms of the scaling parameter $\xi$ we restrict the analysis to the two cases $|\xi| = 0.01$ and $|\xi| = 1.00$, i.e. to the two limiting cases of a significant suppression and no suppression of left-handed interactions between DM and leptons. The value of $y_\psi$ is randomly generated within the range $y_\psi \in [0, 2]$.

The results are shown in Figure 8 and Figure 9, where in the former we have assumed maximum mixing, i.e. we have set $m_\Psi = m_\psi$ and $\theta_\psi = \pi/4$. The DM mass is fixed to $m_{\phi_3} = 600\,\text{Gev}$ and the mass parameters $m_\Psi = m_\psi$ vary. In the limit $m_{\phi_3} \gg m_{\ell_i}$, the bound from the relic density constraint in the SFF scenario reduces to the spherical condition

$$|\lambda_{e3}|^2 + |\lambda_{\mu 3}|^2 + |\lambda_{\tau 3}|^2 \approx \text{const}. \tag{58}$$

---

[10]Note the different definitions of the mass ratio $\mu$, which in Reference [26] is defined as the inverse squared of the definition we use in this work.

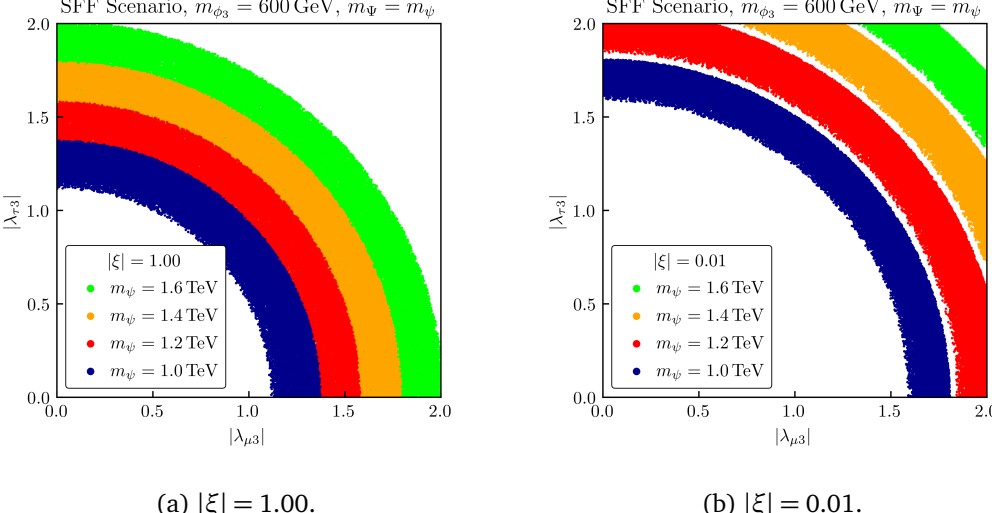

(a) $|\xi| = 1.00$.          (b) $|\xi| = 0.01$.

Figure 8: Constraints on $|\lambda_{\mu 3}|$ and $|\lambda_{\tau 3}|$ from the observed DM relic density in the SFF scenario for maximum mixing with $\theta_\psi = \pi/4$. The DM mass is set to $m_{\phi_3} = 600\,\text{GeV}$ and the mass parameters $m_\Psi = m_\psi$ vary.

This explains the outer edge of the bands that can be seen in Figure 8. The inner edge is due to the flavour constraints, which force the DM–electron coupling $|\lambda_{e3}|$ to be small. We further find that larger couplings $|\lambda_{\mu 3}|$ and $|\lambda_{\tau 3}|$ are necessary in order to comply with the relic density constraint for the case $|\xi| = 0.01$ shown in Figure 8b. This is due to the chirality suppression explained above, which for the case of non-suppressed left-handed interactions shown in Figure 8a is lifted. Thus, in this case the viable couplings are smaller than in the case of suppressed left-handed interactions. Note that the additional annihilation channel into a pair of neutrinos, which does not exist for the model analyzed in [26], only yields subdominant contributions to the annihilation rate for both choices of $|\xi|$. As this annihilation channel is purely governed by left-handed interactions, it is still chirality-suppressed and only contributes to the $p$-wave. Hence, for $|\xi| = 1.00$ this contribution is sub-leading to the $s$-wave contribution of annihilations into a pair of charged leptons given in eq. (54). If on the other hand left-handed interactions are suppressed, i.e. we set $|\xi| = 0.01$, the additional annihilation channel into a pair of neutrinos suffers from a $|\xi|^4$ suppression as it is proportional to the left-handed coupling of DM to leptons. As for the mass dependence with respect to $m_\psi$ or $m_\Psi$, respectively, we find that larger masses require larger couplings for both choices of $|\xi|$, which is due to the $1/m_{\psi_\alpha}^2$ suppression of the $s$-wave contribution $a$ and the $1/m_{\psi_\alpha}^4$ suppression of the $p$-wave contribution $b$ to the annihilation rate. Moving away from maximum mixing, i.e. allowing for different values $m_\Psi \neq m_\psi$ has no qualitative impact on the results. For the case of suppressed left-handed interactions the restrictions in the $|\lambda_{\mu 3}| - |\lambda_{\tau 3}|$ plane trivially only depend on the parameter $m_\psi$, which is the mass parameter of the gauge eigenstate $\psi_2'$ that couples the DM triplet to right-handed leptons. If on the other hand left-handed interactions are not suppressed, choosing different values for $m_\Psi$ and $m_\psi$ increases the mass difference $\Delta m_\psi = m_{\psi_1} - m_{\psi_2}$ which the $s$-wave contribution $a$ depends on, while also increasing the $1/m_{\psi_\alpha}^2$ suppression of $a$. We find that these concurring effects only lead to a very small shift of the contours from Figure 8a to larger couplings, i.e. the increased $1/m_{\psi_\alpha}^2$ suppression dominates over the growth in $\Delta m_\psi$ when choosing $m_\Psi \neq m_\psi$.

We do not show the results of the QDF scenario here as the relevant parameter space in this case is nine-dimensional and the resulting constraints are less apparent. Here, the relic

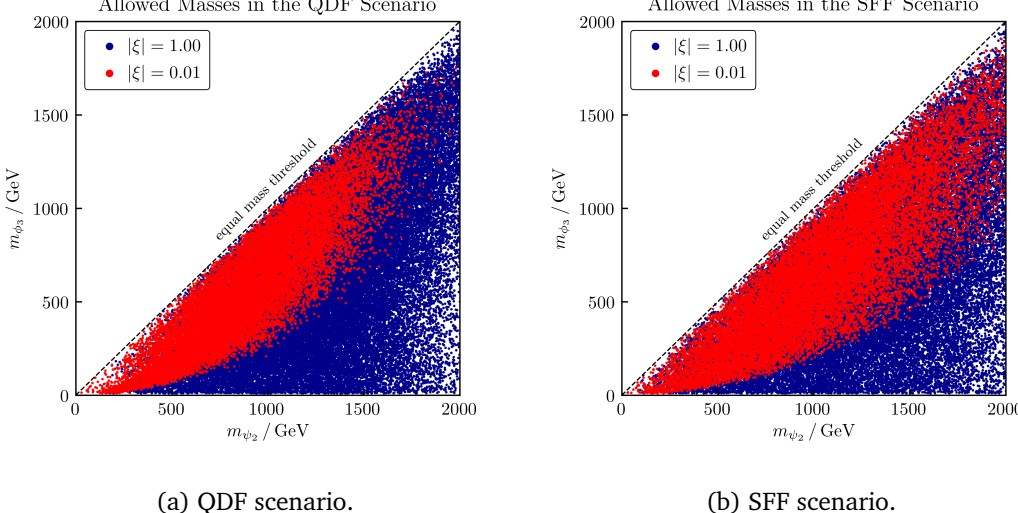

(a) QDF scenario.

(b) SFF scenario.

Figure 9: Allowed masses $m_{\psi_2}$ and $m_{\phi_3}$ for both freeze-out scenarios. The red points correspond to the case of a significant suppression of left-handed interactions and the blue points correspond to the case of no suppression.

density limit reduces for negligible lepton masses to the condition

$$\sum_{ij} |\lambda_{ij}|^2 \approx \text{const.},\tag{59}$$

which corresponds to the shell of a nine-dimensional sphere. Thus, the outer edge of the contours that can be seen in Figure 8 is also present in the QDF scenario, while there is no inner edge due to the sum over initial state flavours. However, the QDF scenario generally requires larger couplings than the SFF scenario in order to satisfy the constraint, since in this case the DM annihilation rate is smaller than in the SFF case due to the flavour-averaging factor in eqs. (45)–(48).

This can also be seen in Figure 9, where we show the allowed masses $m_{\psi_2}$ and $m_{\phi_3}$ for both freeze-out scenarios. For the case of suppressed left-handed interactions between DM and leptons the lower limit on $m_{\phi_3}$ for a given value of $m_{\psi_2}$ is larger in the QDF scenario than in the SFF scenario. This is illustrated by the red points in Figure 9a for the QDF scenario and Figure 9b for the SFF scenario. The leading contribution to the annihilation rate in this case is given by the $p$-wave from eq. (56). For masses $m_\ell \ll m_{\phi_3} \ll m_{\psi_2}$ it behaves like

$$b = \frac{1}{9} \sum_{ij} \sum_{kl} \frac{|\lambda_{ik}|^2 |\lambda_{jl}|^2}{16\pi} \frac{m_{\phi_3}^2}{m_{\psi_1}^2 m_{\psi_2}^2} .\tag{60}$$

As the overall annihilation rate suffers from the above mentioned $p$-wave suppression in this case, growing values for $m_{\psi_2}$ require growing values for $m_{\phi_3}$ in order to not yield a too small annihilation rate or too large relic density, respectively. This explains the lower edge that can be seen for the case $|\xi| = 0.01$ in Figure 9 for both freeze-out scenarios. This lower limit on $m_{\phi_3}$ is larger for the QDF scenario than for the SFF scenario because as mentioned above the overall annihilation rate is smaller in the QDF scenario due to the flavour-average. Hence, in the QDF scenario even larger values of $m_{\phi_3}$ are required in order to yield the correct relic density for a fixed value of $m_{\psi_2}$. For non-suppressed left-handed interactions between DM and leptons the annihilation rate is no longer chirality-suppressed and thus there is no lower limit on $m_{\phi_3}$ in this case. This is shown by the blue points in Figure 9a and Figure 9b and we find

that in this case even very large values for $m_{\psi_2}$ allow for any DM mass $m_{\phi_3} < m_{\psi_2}$. Note that we only show the $m_{\psi_2} - m_{\phi_3}$ plane in Figure 9 as the largest contributions to the annihilation rate come from processes where the light charged mediator $\psi_2$ is exchanged in the $t$-channel. Diagrams with a $\psi_1$-exchange suffer from an additional suppression by a larger NP scale since we conventionally choose $m_{\psi_1} > m_{\psi_2}$.

# 7 DM detection experiments

DM detection and especially direct detection experiments have generally proven to yield strong constraints on flavoured DM models [20–26]. While for lepton-flavoured DM contributions to DM–nucleon scattering are only generated at the one-loop level, in [26] we still found that restrictions coming from direct detection experiments rank among the strongest constraints. There, we had studied a version of this model with purely right-handed interactions between DM and leptons, i.e. the case $\xi = y_\psi = 0$. While indirect detection constraints were found to have a significantly smaller impact on the parameter space of that model, we expect them to gain relevance in this analysis due to our findings from Section 6. As the annihilation rate of DM into SM particles does not necessarily suffer from a chirality suppression in this analysis, relevant contributions to the production rate of electron-positron pairs and photons can become sizeable. We thus use this section to discuss constraints coming from direct and indirect detection experiments.

## 7.1 Direct detection

For the discussion of direct detection constraints we follow the procedure in [26], adopted from [79], and ignore constraints from DM–atom, inelastic DM–electron as well as elastic DM–electron scattering. The former two can be neglected as in these cases DM needs to scatter off bound electrons with a non-negligible momentum of order $p_e \sim \mathcal{O}(1\,\text{MeV})$ in order to generate a sizeable signal. Thus, both processes suffer from a wave-function suppression and can be neglected. The constraints on elastic DM–electron scattering on the other hand, are only relevant for sub-MeV DM [80] that we do not consider in this analysis.

Thus, we are left with DM–nucleon scattering. Relevant contributions to the scattering rate between DM and nucleons arise through the one-loop penguins shown in Figure 10. The process shown in Figure 10a can only be mediated by a photon $\gamma$ if $\alpha = \beta \in \{1, 2\}$, while the case with two neutral mediators in the loop $\alpha = \beta = 0$ is mediated by a $Z$ boson and has a neutrino $\nu_i$ in the loop. Additional diagrams exist for the $Z$ boson mediated case for $\alpha, \beta \in \{1, 2\}$. Since the $Z$ penguin contribution is proportional to the external momentum, its contribution to DM-nucleon scattering can safely be neglected. In the diagram of Figure 10b the indices are restricted to $\alpha, \beta \in \{1, 2\}$. While the diagram where the Higgs boson $h$ is emitted from two charged leptons in the loop is proportional to $y_{\ell_i} y_N |\lambda_{i3}|^2$ and can thus be neglected, the diagram with two charged mediators in the loop is proportional to $y_\psi y_N |\lambda_{i3}|^2$, which can generally become large. Here, $y_N \simeq 0.3$ is the Higgs-nucleon coupling [81]. In fact we find that the latter diagram's amplitude is divergent and contributes to the renormalization of the Higgs-portal coupling $\lambda_{H\phi}$. This coupling gives rise to a tree-level scattering process proportional to $y_N \lambda_{H\phi}$ where DM scatters off a nucleon through a $t$-channel Higgs exchange.[11] We follow the same arguments as in Section 6 and in [25, 26] and use the freedom to choose $\lambda_{H\phi}$ such that the tree-level and one-loop contributions cancel.

This leaves the photon-mediated one-loop penguin from Figure 10a as the only relevant

---

[11]For large parts of the parameter space this contribution is comparable to the photon penguin for $\lambda_{H\phi} \sim \mathcal{O}(1)$ couplings, see Appendix B for details.

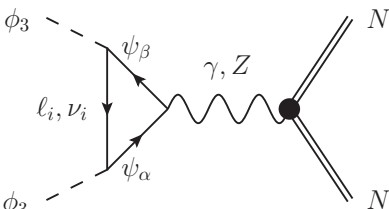

(a) one-loop DM–nucleon scattering mediated by $\gamma$ or $Z$.

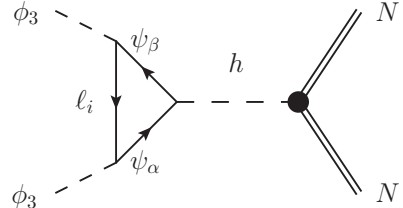

(b) one-loop DM–nucleon scattering mediated by $h$.

Figure 10: Representative Feynman diagrams of relevant interactions for direct detection signals. Note that for both penguins there is also a diagram with two leptons and one mediator in the loop where the bosons are emitted from the two leptons.

contribution to DM–nucleon scattering, which is induced by the charge-radius operator

$$\mathcal{O}_\gamma = \partial^\mu \phi \, \partial^\nu \phi^\dagger F_{\mu\nu}. \tag{61}$$

In the limit of small lepton masses the matched Wilson coefficients $f_{\gamma,\alpha}$ of the contribution with the charged mediator $\psi_\alpha$ in the loop read [27]

$$f_{\gamma,1} = -\sum_i \frac{e\,|\lambda_{i3}|^2\left(s_\theta^2 + |\xi|^2 c_\theta^2\right)}{16\pi^2\, m_{\psi_1}^2}\left[1 + \frac{2}{3}\ln\left(\frac{m_{\ell_i}^2}{m_{\psi_1}^2}\right)\right], \tag{62}$$

$$f_{\gamma,2} = -\sum_i \frac{e\,|\lambda_{i3}|^2\left(c_\theta^2 + |\xi|^2 s_\theta^2\right)}{16\pi^2\, m_{\psi_2}^2}\left[1 + \frac{2}{3}\ln\left(\frac{m_{\ell_i}^2}{m_{\psi_2}^2}\right)\right]. \tag{63}$$

Here we have neglected contributions to $f_{\gamma,\alpha}$ with a chirality flip on the lepton line, as these are suppressed by the negligible lepton masses. In the expressions above the mass $m_e$ needs to be replaced by the momentum transfer $|\vec{q}| = \mathcal{O}(3-10)\,\text{MeV}$ for $i = 1$, i.e. for first generation leptons in the loop [27], as the electron mass is smaller than $|\vec{q}|$.

Using the expressions from above we write for the spin-independent averaged DM–nucleon scattering cross section

$$\sigma_{\text{SI}}^N = \frac{Z^2\, e^2\, \mu^2}{8\pi A^2}\,|f_{\gamma,1} + f_{\gamma,2}|^2\,, \tag{64}$$

where $Z$ and $A$ are the atomic and mass number of the nucleon while $\mu$ is the reduced mass of the DM–nucleon system defined as $\mu = m_N m_{\phi_3}/(m_N + m_{\phi_3})$. In the numerical analysis we use limits obtained from the XENON1T experiment [82] and again use the lepton masses from [52]. The momentum transfer mentioned above is set to $|\vec{q}| = 10\,\text{MeV}$ and for the atomic and mass numbers of Xenon we use $Z = 54$ and $A = 131$, i.e. we ignore the impact of Xenon isotopes as their effect on the overall DM–nucleon scattering cross section was found to be small in [22]. Recall that due to the absence of a flavour symmetry in this model, the mass splittings between the different dark scalars do not depend on the coupling matrix $\lambda$. Hence, the direct detection constraints carry no dependence on the freeze-out scenario.

The results are shown in Figure 11 for maximum mixing between the charged mediators $\psi_1$ and $\psi_2$. The value of the mass parameters $m_\Psi = m_\psi$ varies and the DM mass is fixed to $m_{\phi_3} = 200\,\text{GeV}$. Note that while $\sigma_{\text{SI}}^N$ does not depend on the DM mass, the XENON1T upper limit on the DM–nucleon scattering cross section reaches its minimum at $m_{\phi_3} \simeq 30\,\text{GeV}$ and increases for increasing values of $m_{\phi_3}$. Hence, increasing DM masses generally allow for larger couplings. The same behaviour holds true for increasing values of the charged mediator masses $m_{\psi,\alpha}$, as the amplitudes $f_{\gamma,\alpha}$ are suppressed by $1/m_{\psi_\alpha}^2$. This can be seen in Figure 11

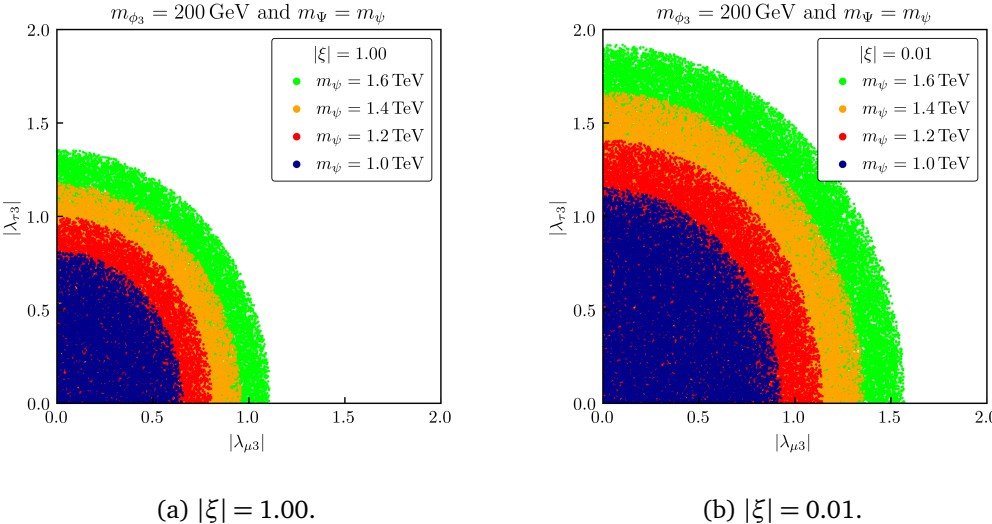

(a) $|\xi| = 1.00$.          (b) $|\xi| = 0.01$.

Figure 11: Allowed couplings $|\lambda_{\mu 3}|$ and $|\lambda_{\tau 3}|$ for both choices of $|\xi|$ and various values for $m_\Psi = m_\psi$, while assuming maximum mixing with $\theta_\psi = \pi/4$. The DM mass is fixed to $m_{\phi_3} = 200$ GeV. The value of $y_\psi$ in randomly generated within the interval $y_\psi \in [0, 2]$.

where we show the distribution of allowed points in the $|\lambda_{\mu 3}| - |\lambda_{\tau 3}|$ plane. For both choices of $|\xi|$ growing mediator masses $m_{\psi_\alpha}$ allow for larger couplings. Trivially, the case of non-suppressed left-handed interactions shown in Figure 11a requires smaller couplings than the suppression case with $|\xi| = 0.01$ shown in Figure 11b, due to additional contributions from left-handed leptons in the loop. In terms of the sizes of $|\lambda_{\mu 3}|$ and $|\lambda_{\tau 3}|$ we find that the DM–tau coupling may grow larger than the DM–muon coupling. This is due to the logarithm of the lepton mass $m_{\ell_i}$ in eqs. (62) and (63). Smaller masses $m_{\ell_i}$ lead to an enhancement of the scattering amplitude $f_{\gamma, \alpha}$ and hence the DM–muon coupling suffers from stronger limits than the DM–tau couplings, due to $m_\mu$ being much smaller than $m_\tau$. We do not show the DM–electron coupling here as we assumed it to be small in order to fulfil the stringent flavour constraints discussed in Section 4.

## 7.2 Indirect detection

We now turn to the discussion of constraints from indirect DM detection experiments. While in our model DM couples to both left- and right-handed leptons, in the case of suppressed left-handed couplings, $|\xi| = 0.01$, the annihilation rate of DM into SM matter still suffers from a chirality suppression. We hence follow our analysis from [26] and include additional diagrams to its calculation in order to lift this suppression. This is necessary in order to properly analyse the indirect detection constraints since the $p$-wave contribution to the annihilation rate suffers from a severe velocity suppression as the DM halo velocity in the Milky Way today is roughly $\langle v^2 \rangle \simeq 10^{-6}$.

The additional diagrams that we consider are shown in Figure 12. The annihilation of two dark scalars $\phi_3$ into a pair of leptons and an additional photon from Figure 12a is referred to as internal bremsstrahlung and lifts the chirality suppression of the annihilation rate in the chiral limit $m_\ell \to 0$ [75]. It is proportional to $\alpha_{\text{em}}/\pi \sim 10^{-3}$ while the box diagram of Figure 12b is even further suppressed by $\alpha_{\text{em}}^2/(4\pi)^2 \sim 10^{-7}$, but gives comparable contributions to the overall annihilation rate in parts of the parameter space. Note that both diagrams are not relevant for the thermal freeze-out, as the DM halo velocity at $T_f$ reads $\langle v^2 \rangle \simeq 0.3$ and hence

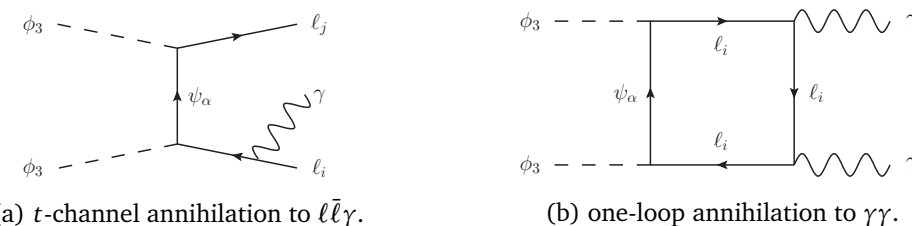

(a) $t$-channel annihilation to $\ell\bar{\ell}\gamma$.

(b) one-loop annihilation to $\gamma\gamma$.

Figure 12: Representative Feynman diagrams for relevant higher-order annihilation processes. The index only refers to charged mediators, i.e. $\alpha \in \{1, 2\}$.

the $p$-wave annihilation into $\bar{\ell}_i \ell_j$ is much less suppressed than today.

The annihilation rate for the process of Figure 12b reads

$$\langle\sigma v\rangle_{\gamma\gamma} = \langle\sigma v\rangle_{\gamma\gamma}^1 + \langle\sigma v\rangle_{\gamma\gamma}^2 + 2\langle\sigma v\rangle_{\gamma\gamma}^{12}, \tag{65}$$

where the superscript denotes the contributions from diagrams where either $\psi_1$ or $\psi_2$ is exchanged in the loop as well as the interference term. For the same reason as for the one-loop photon penguin in Figure 10a the contributions with a chirality flip on a lepton line vanish in the chiral limit $m_\ell \to 0$ [83]. In this limit, the expressions from eq. (65) are given by [83]

$$\langle\sigma v\rangle_{\gamma\gamma}^1 = \frac{\alpha_{\text{em}}^2 \left(s_\theta^2 + |\xi|^2 c_\theta^2\right)^2}{64\pi^3 m_{\phi_3}^2} \left(\sum_i |\lambda_{i3}|^2\right)^2 |\mathcal{B}(\mu_1)|^2, \tag{66}$$

$$\langle\sigma v\rangle_{\gamma\gamma}^2 = \frac{\alpha_{\text{em}}^2 \left(c_\theta^2 + |\xi|^2 s_\theta^2\right)^2}{64\pi^3 m_{\phi_3}^2} \left(\sum_i |\lambda_{i3}|^2\right)^2 |\mathcal{B}(\mu_2)|^2, \tag{67}$$

$$\langle\sigma v\rangle_{\gamma\gamma}^{12} = \frac{\alpha_{\text{em}}^2 \left(c_\theta^2 + |\xi|^2 s_\theta^2\right)\left(s_\theta^2 + |\xi|^2 c_\theta^2\right)}{64\pi^3 m_{\phi_3}^2} \left(\sum_i |\lambda_{i3}|^2\right)^2 |\mathcal{B}(\sqrt{\mu_1\mu_2})|^2. \tag{68}$$

The loop function $\mathcal{B}$ is defined as

$$\mathcal{B}(\mu_\alpha) = 2 - 2\log\left[1 - \frac{1}{\mu_\alpha}\right] - 2\mu_\alpha \arcsin\left[\frac{1}{\sqrt{\mu_\alpha}}\right]^2, \tag{69}$$

where $\mu_\alpha = \psi_\alpha^2 / m_{\phi_3}^2$.

Similarly, we decompose the annihilation rate into the three-body final state of Figure 12a and write

$$\langle\sigma v\rangle_{\ell\bar{\ell}\gamma} = \langle\sigma v\rangle_{\ell\bar{\ell}\gamma}^1 + \langle\sigma v\rangle_{\ell\bar{\ell}\gamma}^2 + 2\langle\sigma v\rangle_{\ell\bar{\ell}\gamma}^{12}. \tag{70}$$

For this process, the contributions with a chirality flip on any of the external fermion lines vanish in the limit of zero lepton masses. On the other hand, diagrams with a chirality flip on the virtual mediator in the $t$-channel only lead to $p$-wave suppressed and therefore negligible contributions.[12] However, the calculation of the interference term between the two diagrams with either a $\psi_1$ or a $\psi_2$ in the $t$-channel is less trivial than for $\langle\sigma v\rangle_{\gamma\gamma}$ due to the three-body phase space. Following the procedure presented in [86,87] we have obtained an expression for $\langle\sigma v\rangle_{\ell\bar{\ell}\gamma}^{12}$ that can be found in Appendix C and was tested to yield the correct total annihilation rate $\langle\sigma v\rangle_{\ell\bar{\ell}\gamma}$ in the limit $|\xi| = y_\psi = 0$. The other two contributions read [75,88]

$$\langle\sigma v\rangle_{\ell\bar{\ell}\gamma}^1 = \frac{\alpha_{\text{em}} \left(s_\theta^2 + |\xi|^2 c_\theta^2\right)^2}{32\pi^2 m_{\phi_3}^2} \sum_{ij} |\lambda_{i3}|^2 |\lambda_{j3}|^2 \mathcal{A}(\mu_1), \tag{71}$$

---

[12]This $p$-wave suppression is due to the conservation of the total angular momentum, since the annihilation of two scalars in the $s$-wave implies $J = 0$ while the photon only has two polarizations with $J_z \in \{-1, 1\}$. Note that the same finding also holds true for the annihilation of neutralinos into a pair of fermions and a photon, see [84,85].

$$\langle\sigma v\rangle^2_{\ell\bar{\ell}\gamma} = \frac{\alpha_{\text{em}}\left(c_\theta^2 + |\xi|^2 s_\theta^2\right)^2}{32\pi^2 m_{\phi_3}^2} \sum_{ij} |\lambda_{i3}|^2 |\lambda_{j3}|^2 \mathcal{A}(\mu_2),\tag{72}$$

and the phase space function $\mathcal{A}(\mu_\alpha)$ is defined according to

$$\mathcal{A}(\mu_\alpha) = (\mu_\alpha + 1)\left(\frac{\pi^2}{6} - \log^2\left[\frac{\mu_\alpha + 1}{2\mu_\alpha}\right] - 2\text{Li}_2\left[\frac{\mu_\alpha + 1}{2\mu_\alpha}\right]\right)$$
$$+ \frac{4\mu_\alpha + 3}{\mu_\alpha + 1} + \frac{4\mu_\alpha^2 - 3\mu_\alpha - 1}{2\mu_\alpha}\log\left[\frac{\mu_\alpha - 1}{\mu_\alpha + 1}\right].\tag{73}$$

Here, $\text{Li}_2(z)$ is the dilogarithm and we have again used $\mu_\alpha = m_{\psi_\alpha}^2/m_{\phi_3}^2$.

Last but not least, the tree-level rate of DM annihilating into a pair of leptons $\bar{\ell}_i\ell_j$ is given by the expression for the SFF scenario's thermal annihilation rate from Section 6.

To study the indirect detection constraints numerically we use limits obtained from the AMS experiment [89] and from measurements by the Fermi-LAT satellite [90] as well as the H.E.S.S. telescope [91]. Reference [92] calculated an upper limit $\langle\sigma v\rangle^{\text{max}}_{\bar{e}}$ on the annihilation rate of Majorana DM into an electron-positron pair with a branching fraction of 100% based on AMS-02 measurements of the positron flux. While this signal generally includes prompt as well as secondary positrons stemming from decays of heavy charged leptons, the energy spectrum of the latter is shifted towards lower energies compared to prompt positrons. Further, secondary positrons additionally suffer from a smeared momentum distribution so that the AMS-02 limit $\langle\sigma v\rangle^{\text{max}}_{\bar{e}}$ mainly constrains prompt positrons. We thus sum over the annihilation rates of all processes with a positron in the final state and compare with the experimental upper limit. Here we follow our analysis in [26] and also include the radiative corrections shown in Figure 12a, i.e. we compare the sum

$$\langle\sigma v\rangle_{\bar{e}} = \sum_\ell \langle\sigma v\rangle_{\ell\bar{e}} + \langle\sigma v\rangle_{\ell\bar{e}\gamma},\tag{74}$$

with the upper limit $\langle\sigma v\rangle^{\text{max}}_{\bar{e}}$. In doing so we ignore the shift in the $m_{\phi_3}$ dependence of the three-body final state with comparison to the two-body final state, as we assume it to be small.

Using measurements of the $\gamma$-ray continuum spectrum by Fermi-LAT, Reference [93] provides an equivalent limit $\langle\sigma v\rangle^{\text{max}}_\tau$ on the annihilation of Majorana DM into a tau-antitau pair. Just as the constraints from the positron flux are most sensitive to prompt signals, this upper limit is dominated by annihilations into taus, as such final states produce more photons through subsequent decays than final states with muons or electrons. Hence we only focus on final states with at least one tau or antitau and compare the total annihilation rate with the experimental upper limit $\langle\sigma v\rangle^{\text{max}}_\tau$. To this end we define

$$\langle\sigma v\rangle_\tau = \langle\sigma v\rangle_{\tau\bar{\tau}} + \langle\sigma v\rangle_{\tau\bar{\tau}\gamma} + \frac{1}{2}\sum_{\ell=e,\mu}\left(\langle\sigma v\rangle_{\ell\bar{\tau}} + \langle\sigma v\rangle_{\bar{\ell}\tau} + \langle\sigma v\rangle_{\ell\bar{\tau}\gamma} + \langle\sigma v\rangle_{\bar{\ell}\tau\gamma}\right),\tag{75}$$

which in total gives five annihilation channels with a tau or antitau in the final state and five radiative corrections, respectively. Here we have included a factor of 1/2 for final states with a single tau or antitau, since $\langle\sigma v\rangle^{\text{max}}_\tau$ was derived for the final state consisting of a tau-antitau pair.

Finally, we also consider indirect detection limits obtained by Fermi-LAT and H.E.S.S. measurements of the $\gamma$-ray line spectrum. The energy spectrum of $\gamma$-rays produced by the process shown in the box diagram of Figure 12b trivially peaks at the energy $E_\gamma = m_{\phi_3}$ due to energy-momentum conservation. The internal bremsstrahlung process on the other hand is dominated by hard photons [94] with $E_\gamma \approx m_{\phi_3}$ emitted by the charged mediator $\psi_\alpha$. This process is referred to as virtual internal bremsstrahlung and exhibits a line-like $\gamma$-ray energy spectrum with

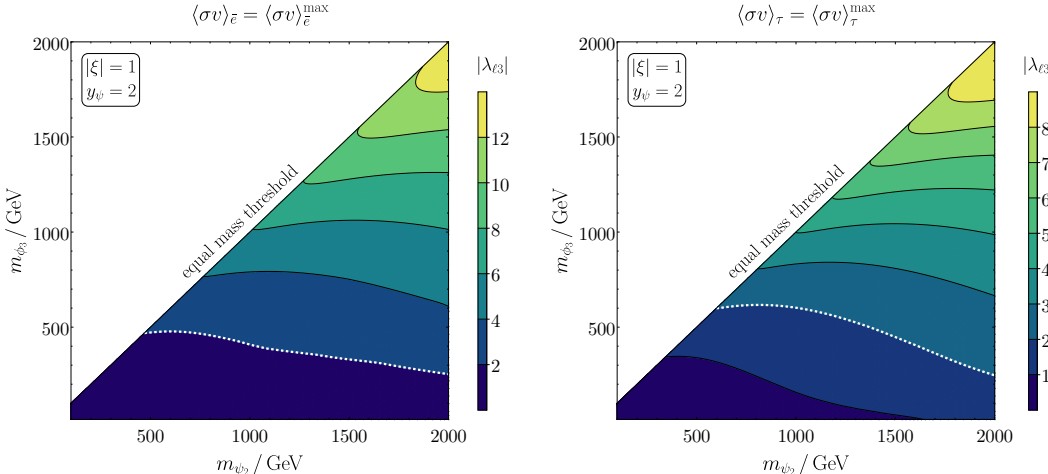

(a) constraints from the positron flux.

(b) constraints from the $\gamma$ continuum spectrum.

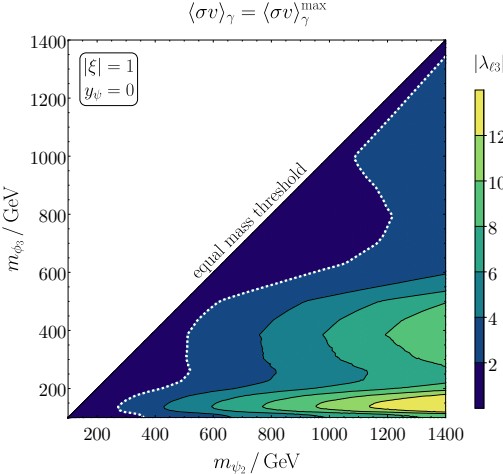

(c) constraints from the $\gamma$ line spectrum.

Figure 13: Restrictions on the model parameters from indirect detection experiments for non-suppressed left-handed interactions. In all three plots we assume a maximum mixing with $\theta_\psi = \pi/4$. The area included by the white dashed line, the horizontal axis and the equal mass diagonal indicates in which mass regime the constraints are relevant.

a sharp peak at energies slightly below the DM mass [90, 94]. Reference [94] provided a limit $\langle\sigma v\rangle_\gamma^{\max}$ based on searches for such lines in the $\gamma$-ray spectrum performed by the Fermi-LAT satellite and the H.E.S.S. telescope which is derived for the annihilation rate

$$\langle\sigma v\rangle_\gamma = \sum_\ell \langle\sigma v\rangle_{\ell\bar\ell\gamma} + 2\langle\sigma v\rangle_{\gamma\gamma}. \tag{76}$$

We use these expressions in our numerical analysis of indirect detection constraints to further restrict the parameter space of our model.

In order to identify the mass region in the $m_{\psi_2} - m_{\phi_3}$ plane in which indirect detection constraints are relevant, we set the DM–lepton couplings to a universal value $|\lambda_{i3}| = |\lambda_{\ell 3}|$ and check how large it may maximally grow. To this end we allow the annihilation rates $\langle\sigma v\rangle_{\bar e}$, $\langle\sigma v\rangle_\tau$ and $\langle\sigma v\rangle_\gamma$ to grow as large as the respective experimental upper limit. The resulting contours are shown in Figure 13 for all three constrained annihilation rates. We

show the case of non-suppressed left-handed interactions between DM and leptons and maximum mixing between $\psi_1$ and $\psi_2$. The annihilation rates $\langle\sigma v\rangle_{\bar{e}}$ and $\langle\sigma v\rangle_{\tau}$ are dominated by the $s$-wave annihilation from eq. (54), which is proportional to the mass difference $\Delta m_{\psi} = m_{\psi_1} - m_{\psi_l 2} = \sqrt{2}y_{\psi}v$. Hence, we set $y_{\psi} = 2.0$ in Figure 13a and 13b to determine the largest possible constraints. The annihilation rate $\langle\sigma v\rangle_{\gamma}$ on the other hand, does not depend on $\Delta m_{\psi}$ and grows for decreasing values of $y_{\psi}$ and a fixed mediator mass $m_{\psi_2}$, since decreasing Yukawa couplings $y_{\psi}$ reduce the mass $m_{\psi_1}$. This leads to larger contributions to $\langle\sigma v\rangle_{\gamma}$ from the diagrams of Figure 12 with $\psi_1$ in the $t$-channel or loop, respectively. In all three figures the white dashed line indicates where the respective constraint forces the DM–lepton coupling to be $|\lambda_{\ell 3}| \leq 2.0$, i.e. constraints are only relevant in the areas included by the horizontal axis, the equal mass threshold and this line. We find that the indirect detection constraints are dominated by limits obtained from measurements of the $\gamma$-ray spectrum shown in Figure 13b and Figure 13c. The former limits are relevant for DM masses $m_{\phi_3} \lesssim 600\,\text{GeV}$ and over the complete range of $m_{\psi_2}$, while searches for line features of the $\gamma$-ray spectrum generally become relevant close to the equal mass threshold. For mediator masses $m_{\psi_2} \lesssim 1200\,\text{GeV}$ this limit can also become relevant for larger mass splittings between $m_{\psi_2}$ and $m_{\phi_3}$. We do not show the case of suppressed left-handed interactions with $|\xi| = 0.01$ here but relegate it to Appendix C, since this case yields exactly the same contours as for the purely right-handed version of this model studied by us in [26]. There we found the constraints from the positron flux as well as the continuum $\gamma$-ray spectrum to be much weaker, due to the chirality suppression of the annihilation rate into $\ell_i \bar{\ell}_j$ mentioned above. In total we conclude that in spite of yielding much stronger restrictions on the parameter space for the case of unsuppressed left-handed couplings, $|\xi| = 1.00$, the indirect detection constraints are weaker than limits from direct detection, LFV or the DM relic density.

## 8   Combined analysis

In order to obtain a global picture of the viable parameter space of our model we use this section to perform a combined analysis of all constraints discussed in the previous sections. To do so, we generate random points in the parameter space and demand that all constraints are simultaneously fulfilled. The results of this combined numerical analysis are gathered in Figure 14, Figure 15 and Figure 16.

Figure 14 shows the viable parameter space in the $m_{\psi_2} - m_{\phi_3}$ plane for both freeze-out scenarios and both cases of $|\xi|$. We further show the largest possible exclusion[13] in this plane stemming from the LHC searches discussed in Section 3 and find that for both choices of $|\xi|$ they only lead to an additional exclusion for the QDF scenario. For the SFF scenario we find that the allowed masses are roughly the same for both choices of $|\xi|$ and are mainly determined by the interplay of the relic density and direct detection constraints. While the annihilation rate for the case $|\xi| = 1.00$ is not chirality suppressed, it still suffers from a suppression by $\Delta m_{\psi}^2/(m_{\psi_1} m_{\psi_2})$ with $\Delta m_{\psi} = m_{\psi_1} - m_{\psi_2} = \sqrt{(m_{\Psi} - m_{\psi})^2 + 2y_{\psi}^2 v^2}$. Thus, either large couplings or large DM masses are needed in order to push the annihilation rate high enough to not yield over-abundant DM. At the same time, large couplings are subject to strong constraints from direct detection experiments and thus both constraints can only be fulfilled for masses $m_{\psi_2} \gtrsim 800\,\text{GeV}$ and $m_{\phi_3} \gtrsim 600\,\text{GeV}$. In these ranges, either the DM annihilation rate is sufficiently enhanced by the DM mass $m_{\phi_3}$, such that couplings small enough to pass the direct

---

[13]Note that strictly speaking these limits do not straightforwardly apply here, since they assume $e-\mu$ universality while we have fixed the DM–electron couplings to $|\lambda_{ei}| \in [10^{-6}, 10^{-1}]$ in the combined analysis. We hence expect the actual exclusion from LHC searches to be smaller than the curve shown in Figure 14 and only include it here for illustration purposes.

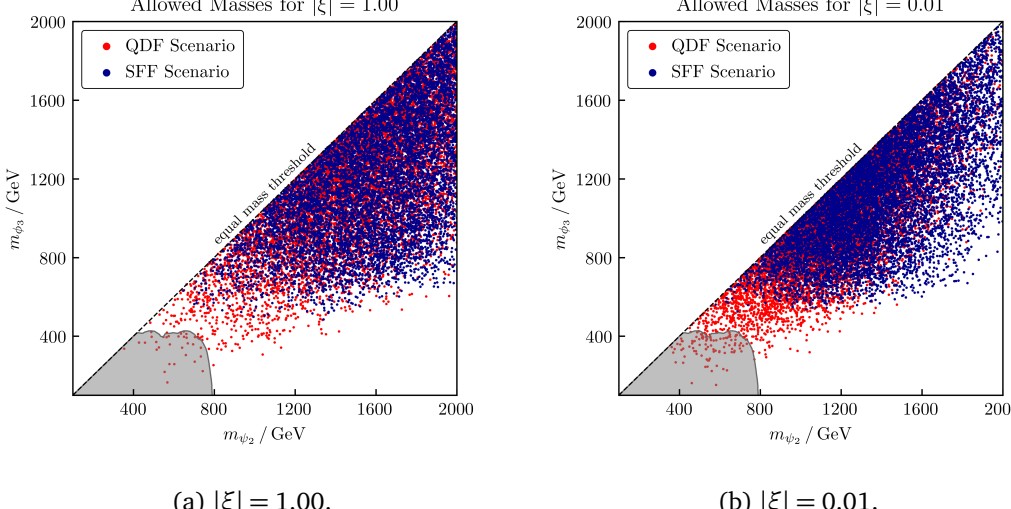

(a) $|\xi| = 1.00$.     (b) $|\xi| = 0.01$.

Figure 14: Allowed masses $m_{\psi_2}$ and $m_{\phi_3}$ while satisfying all constraints. We show both freeze-out scenarios and both cases of $|\xi|$. The grey area shows the largest possible exclusion from LHC searches for same-flavour final states $\ell\bar{\ell} + \not{E}_T$ with $\ell = e, \mu$ discussed in Section 3 and corresponds to the case $|\lambda_{\ell 3}| = 2$, $|\lambda_{\tau 3}| = 0$ and $y_\psi = 0.25$.

detection constraint are viable, or the DM–nucleon scattering rate is sufficiently suppressed by the mediator mass $m_{\psi_2}$ such that large couplings necessary for the correct relic density are allowed. In the QDF scenario the relic density constraint can in principle be fulfilled by annihilations of the heavier dark species $\phi_1$ and $\phi_2$ alone, while the direct detection constraints only restrict the couplings of the lightest state $\phi_3$. Hence, small mediator masses $m_{\psi_2} \lesssim 800\,\text{GeV}$ also become viable for this freeze-out scenario. However, we still encounter a lower limit on $m_{\phi_3}$ for both cases of $|\xi|$. For non-suppressed left-handed interactions this limit is due to the suppression of the $s$-wave annihilation proportional to $v^2/(m_{\psi_1}m_{\psi_2})$. Growing values of $m_{\psi_\alpha}$ further suppress the annihilation rate and thus one needs correspondingly growing DM masses $m_{\phi_3}$ in order to compensate for this suppression. Such a lower limit does not arise from the relic density constraint alone, as the direct detection constraints force the couplings of $\phi_3$ to the SM to be small and therefore significantly reduce the overall annihilation rate. For $|\xi| = 0.01$ on the other hand, we re-encounter the lower limit on $m_{\phi_3}$ from Figure 9, which was due to the chirality suppression of the annihilation rate and the accompanying velocity suppression of the $p$-wave. This again gives rise to a lower limit on $m_{\phi_3}$ in order to compensate for this suppression. Note that for both freeze-out scenarios and both choices of $|\xi|$, growing values of $m_{\phi_3}$ do not only enhance the annihilation rate but also reduce the relevance of the direct detection constraint as the XENON1T upper limit grows for increasing DM masses.

 Figure 15 shows the implications of our combined analysis for the couplings $|\lambda_{\tau 3}|$ and $|\lambda_{\mu 3}|$ in the SFF scenario, for both choices of $|\xi|$. The overall picture is dominated by the relic density, flavour and direct detection constraints. Since, in order to suppress the LFV constraints studied in Section 4, we chose the DM–electron coupling $|\lambda_{e3}|$ to be small, the couplings $|\lambda_{\tau 3}|$ and $|\lambda_{\mu 3}|$ have to be large according to the spherical condition

$$|\lambda_{e3}|^2 + |\lambda_{\mu 3}|^2 + |\lambda_{\tau 3}|^2 \approx \text{const.}, \tag{77}$$

which needs to be satisfied in order to yield the correct relic density. Thus, the interplay of these two constraints leads to the bands of Figure 15, where the flavour constraints cause their inner edge due to the suppressed DM–electron coupling. The outer edge of the bands

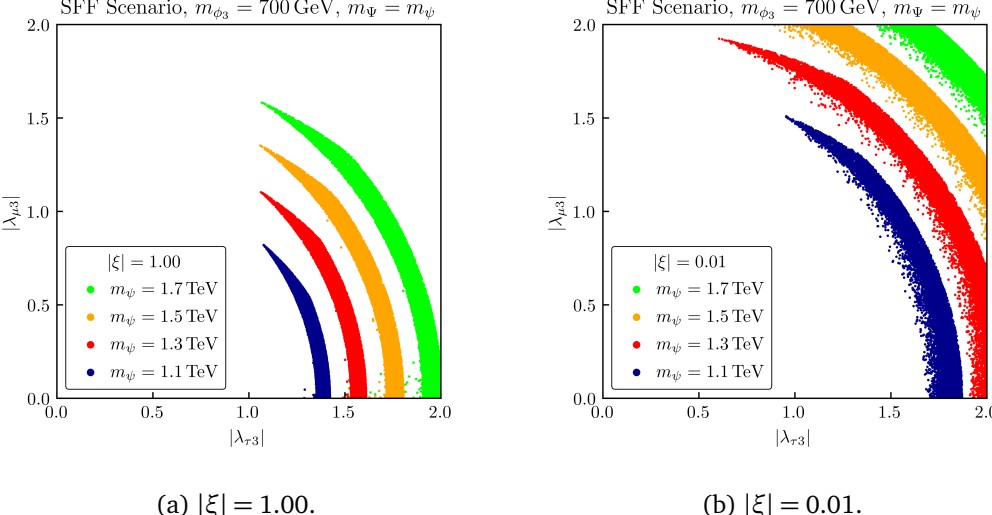

(a) $|\xi| = 1.00$.                    (b) $|\xi| = 0.01$.

Figure 15: Allowed couplings $|\lambda_{\tau 3}|$ and $|\lambda_{\mu 3}|$ while satisfying all constraints in the SFF scenario for both cases of $|\xi|$. We assume maximum mixing with $\theta_\psi = \pi/4$. The DM mass is fixed to $m_{\phi_3} = 700\,\text{GeV}$ and the mass parameters $m_\Psi = m_\psi$ vary.

on the other hand, is mainly determined by the relic density constraint for sufficiently small couplings $|\lambda_{\mu 3}|$. Once an $m_{\psi_2}$-dependent threshold is exceeded with respect to the value of $|\lambda_{\mu 3}|$, the direct detection constraint starts to dominate over the relic density limit, giving rise to the spikes at the upper end of the bands. The direct detection constraint only dominates for large $|\lambda_{\mu 3}|$ as it is more stringent for light leptons in the loop. This is due to the logarithm of the mass $m_{\ell_i}$ in eqs. (62) and (63). In terms of the scaling parameter $|\xi|$ we find that the case of suppressed left-handed interactions shown in Figure 15b allows for larger couplings $|\lambda_{\tau 3}|$ and $|\lambda_{\mu 3}|$. This is due to softened restrictions from direct detection together with the above-mentioned chirality suppression of the annihilation rate in this case and has important implications for the flavour of $\phi_3$. For the case $|\xi| = 0.01$ we find that $\mu$-flavoured DM is viable for masses $m_\psi \gtrsim 1000\,\text{GeV}$. DM with a predominant coupling to the muon is even equally favoured as $\tau$-flavoured DM for masses $m_\psi \gtrsim 1400\,\text{GeV}$. If left-handed interactions between DM and leptons are not suppressed, on the other hand, we find that large parts of the viable parameter space correspond to $\tau$-flavoured DM. Here, $\mu$-flavoured DM only becomes viable in a tiny part of the parameter space for masses $m_\psi \gtrsim 1300\,\text{GeV}$. In both cases $e$-flavoured DM is excluded by our choice to accomodate the strong flavour constraints by suppressing the DM–electron coupling. The latter is necessary in order to be able to obtain a large contribution to the anomalous magnetic moment of the muon and thereby solve the $(g-2)$ anomaly.

The effects of the combined analysis on the $|\lambda_{\tau 3}| - |\lambda_{\mu 3}|$ plane for the QDF scenario are shown in Figure 16. Here, the direct detection constraint dominates for large parts of the parameter space, as the correct relic density can generally also be obtained through annihilations of the heavier states $\phi_1$ and $\phi_2$ alone. This is especially the case for $|\xi| = 1.00$ which we show in Figure 16a and where the direct detection constraint dominates for each choice of $m_\Psi = m_\psi$. While this also holds true for large parts of the parameter space for the case of suppressed left-handed interactions shown in Figure 16b, we here find that for large mediator masses $m_\psi \gtrsim 1600\,\text{GeV}$ the relic density constraint yields a lower limit on the couplings $|\lambda_{\tau 3}|$ and $|\lambda_{\mu 3}|$. As for the flavour of $\phi_3$, we find that the QDF scenario allows for both $\mu$- and $\tau$-flavoured DM. Here, the latter case is slightly favoured over the former, due to stronger direct detection constraints for DM coupling predominantly to muons.

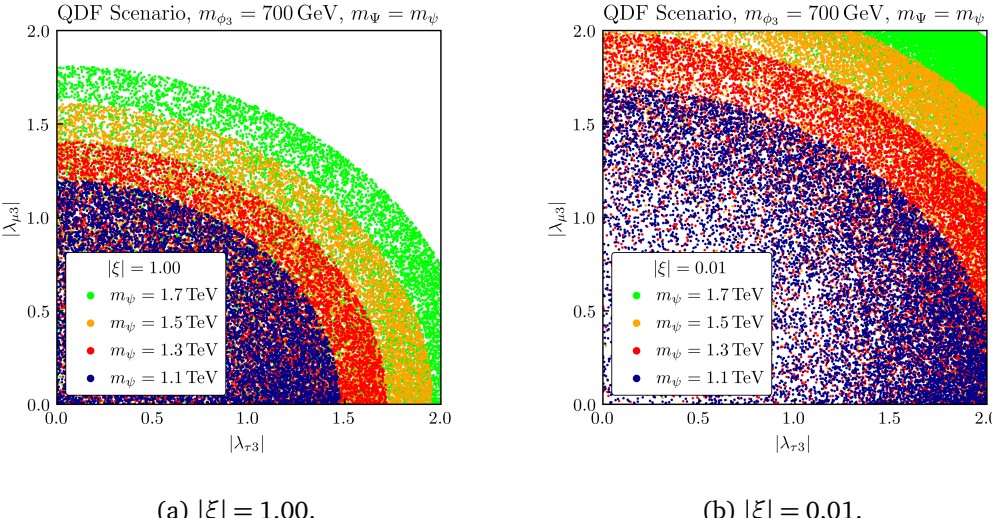

(a) $|\xi| = 1.00$.

(b) $|\xi| = 0.01$.

Figure 16: Allowed couplings $|\lambda_{\tau 3}|$ and $|\lambda_{\mu 3}|$ while satisfying all constraints in the QDF scenario for both cases of $|\xi|$. We assume maximum mixing with $\theta_\psi = \pi/4$. The DM mass is fixed to $m_{\phi_3} = 700\,\text{GeV}$ and the mass parameters $m_\Psi = m_\psi$ vary.

## 9 Muon anomalous magnetic moment

As already mentioned in the introduction we propose this model as a joint solution for the DM problem and the long-standing discrepancy between experimental measurements and the theory prediction of the muon anomalous magnetic moment $a_\mu$. After having identified viable regions of the parameter space of our model, we are now prepared to determine if sizeable NP contributions to $a_\mu$ can be generated.

### 9.1 Theoretical approach

Precision measurements of the muon anomalous magnetic moment [2, 95, 96] yield a world average of

$$a_\mu^{\text{exp}} = (116592059 \pm 22) \times 10^{-11}, \tag{78}$$

while state-of-the-art SM calculations [97–116] predict the value [3]

$$a_\mu^{\text{SM}} = (116591810 \pm 43) \times 10^{-11}. \tag{79}$$

The difference between the theory prediction and measurement reads

$$\Delta a_\mu^{\text{exp}} = a_\mu^{\text{exp}} - a_\mu^{\text{SM}} = (2.49 \pm 0.48) \times 10^{-9}, \tag{80}$$

and corresponds to a significance of $5.1\sigma$.[14] We interpret this tension between theory and experiment as a NP contribution potentially originating from our model.

Such contributions $\Delta a_\mu$ are generated through the diagram shown in Figure 4 with $\ell_i = \ell_j = \mu$ and they read

$$\Delta a_\mu = \Delta a_\mu^1 + \Delta a_\mu^2, \tag{81}$$

---

[14]Using recent lattice determinations of the hadronic vacuum polarisation significantly softens the tension between data and the SM [117–120]. However, in that case, a tension emerges in low-energy $\sigma(e^+e^- \to \text{hadrons})$ data [121–123] that requires further investigation. In the present paper, we hence disregard the lattice results and consider the discrepancy as given in eq. (80).

where the expressions for $\Delta a_\mu^\alpha$ are given in eqs. (34) and (35). Due to the chirality-flipping nature of the anomalous magnetic moment, NP contributions with a chirality flip inside the loop can receive a strong enhancement. In our model, its source is the Yukawa coupling $y_\psi$, which couples the fields $\Psi$ and $\psi_2'$ to the SM Higgs doublet and induces a mixing between the two charged mediators. Thus, in the limit of approximately equal mediator mass parameters $m_\Psi \approx m_\psi$, the scale of the relevant NP giving rise to the chirality flip is given by the mass splitting

$$\Delta m_\psi = m_{\psi_1} - m_{\psi_2} = \sqrt{2} y_\psi v \,, \tag{82}$$

and satisfies $\Delta m_\psi \gg m_\mu$ if $y_\psi \gtrsim 10^{-3}$.

Neglecting hence the first terms of eqs. (34) and (35) and writing

$$\Delta a_\mu = \frac{m_\mu}{16\pi^2} \sum_k \frac{\sin 2\theta_\psi \, |\lambda_{\mu k}|^2}{3 m_{\phi_k}^2} \operatorname{Re} \xi \left( m_{\psi_1} G(x_{k,1}) - m_{\psi_2} G(x_{k,2}) \right), \tag{83}$$

gives a very good approximation of the NP contributions to $a_\mu$. Note that $\theta_\psi$ as defined in eq. (9) lies within $0 \le \theta_\psi \le \pi/4$ such that $\sin 2\theta_\psi > 0$ in the equation above. As we additionally have

$$m_{\psi_2} G(x_{k,2}) > m_{\psi_1} G(x_{k,1}), \tag{84}$$

a positive NP contribution $\Delta a_\mu$ requires a negative scaling parameter, $\xi < 0$.

At the same time the Yukawa coupling $y_\psi$ also generates potentially sizeable NP contributions $\Delta m_\mu$ to the muon mass through the processes depicted in Figure 17. The total muon mass is given by the relation

$$m_\mu = \frac{y_\mu v}{\sqrt{2}} + \Delta m_\mu \,, \tag{85}$$

inducing a potential fine-tuning problem. A general parametric estimate of the NP contributions to $a_\mu$ and $m_\mu$ gives [124, 125]

$$\Delta a_\mu = \mathcal{C}_{\text{NP}} \frac{m_\mu^2}{m_{\text{NP}}^2} \,, \tag{86}$$

$$\Delta m_\mu = \mathcal{O}(\mathcal{C}_{\text{NP}}) m_\mu \,, \tag{87}$$

where the factor $\mathcal{C}_{\text{NP}}$ depends on the details of the model. These expressions can be used in order to derive an upper limit on the NP scale $m_{\text{NP}}$ up to which the experimental value $\Delta a_\mu^{\text{exp}}$ can be accommodated without introducing fine-tuning. To this end, we follow the convention from [124] and consider scenarios in which corrections to the muon mass are larger than the physical muon mass as fine-tuned, which yields an upper limit of [124]

$$m_{\text{NP}} \lesssim 2100 \, \text{GeV} \,. \tag{88}$$

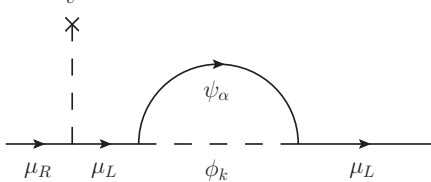

(a) chirality flip on external muon line.

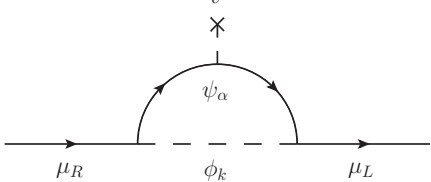

(b) chirality flip in the loop.

Figure 17: Representative Feynman diagrams for NP contributions $\Delta m_\mu$ to the muon mass.

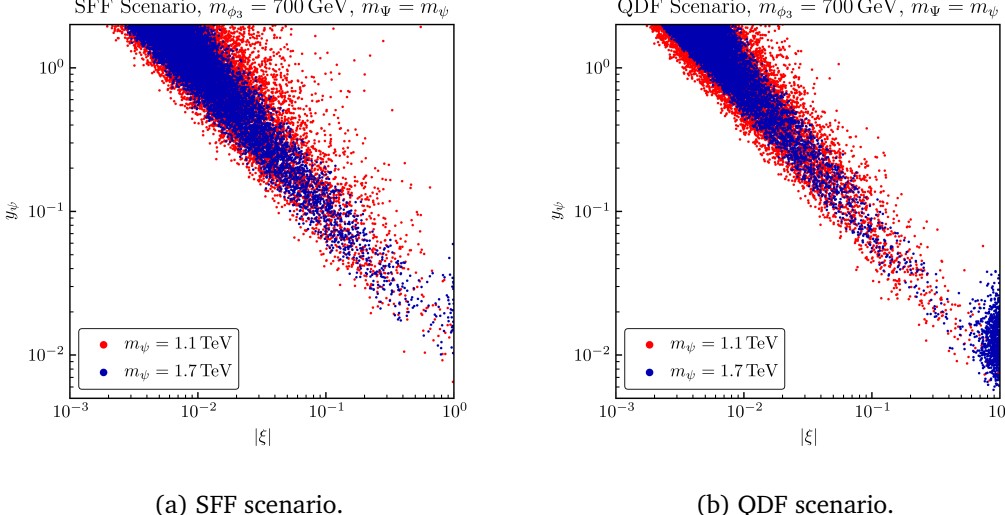

(a) SFF scenario.          (b) QDF scenario.

Figure 18: Viable points in the $|\xi| - y_\psi$ plane while demanding that $\Delta a_\mu$ lies within the $2\sigma$ band of $\Delta a_\mu^{\text{exp}}$.

To complement this general estimate we also check numerically which parts of our model's viable parameter space correspond to fine-tuned scenarios by calculating $\Delta m_\mu$ through [125]

$$\Delta m_\mu = -\frac{\sin 2\theta_\psi \operatorname{Re} \xi}{16\pi^2} \sum_k |\lambda_{\mu k}|^2 \left( m_{\psi_1} B_0(0, m_{\psi_1}, m_{\phi_k}) - m_{\psi_2} B_0(0, m_{\psi_2}, m_{\phi_k}) \right), \qquad (89)$$

where the function $B_0(p^2, m_1, m_2)$ is a standard Passarino–Veltman two-point function renormalised according to the $\overline{\text{MS}}$ scheme. Here we only consider contributions to $\Delta m_\mu$ from the process with a chirality flip in the loop depicted in Figure 17b. The process with a chirality flip on an external muon line shown in Figure 17a is proportional to $m_\mu$ and can hence be safely neglected.

In passing we note that one-loop contributions to the lepton Yukawa couplings are generated by diagrams analogous to Figure 17. The leading contribution, however, affects the lepton mass and the respective Yukawa coupling equally and therefore does not modify the Higgs decay rates to leptons. Corrections to the latter receive an additional $v^2/m_{\psi_\alpha}^2$ suppression factor and are hence smaller than the LHC sensitivity. These conclusions agree with the findings of [32].

## 9.2 Results

In order to determine the size of our model's contributions to $a_\mu$, we calculate $\Delta a_\mu$ in the regions of its parameter space that we have identified as viable in the combined analysis of Section 8. For the calculation of $\Delta a_\mu$ we use the full expression from eq. (81) including diagrams with chirality flips on external muon lines. The NP contributions to the muon mass are calculated through eq. (89). The results are gathered in Figure 18–20. In all plots we assume maximum mixing with $\theta_\psi = \pi/4$.

Figure 18 shows which values of the Yukawa coupling $y_\psi$ and the scaling parameter $|\xi|$ can solve the $(g-2)_\mu$ anomaly in the two freeze-out scenarios. We find that for the case $|\xi| = 1.00$ one needs small Yukawa couplings $y_\psi \lesssim 10^{-1}$ in order to stay within the $2\sigma$ band of $\Delta a_\mu^{\text{exp}}$, while suppressed left-handed interactions with $|\xi| = 0.01$ require $y_\psi \gtrsim 0.3$, with values as large as $y_\psi = 2.0$ possible. Comparing the two freeze-out scenarios, we find that

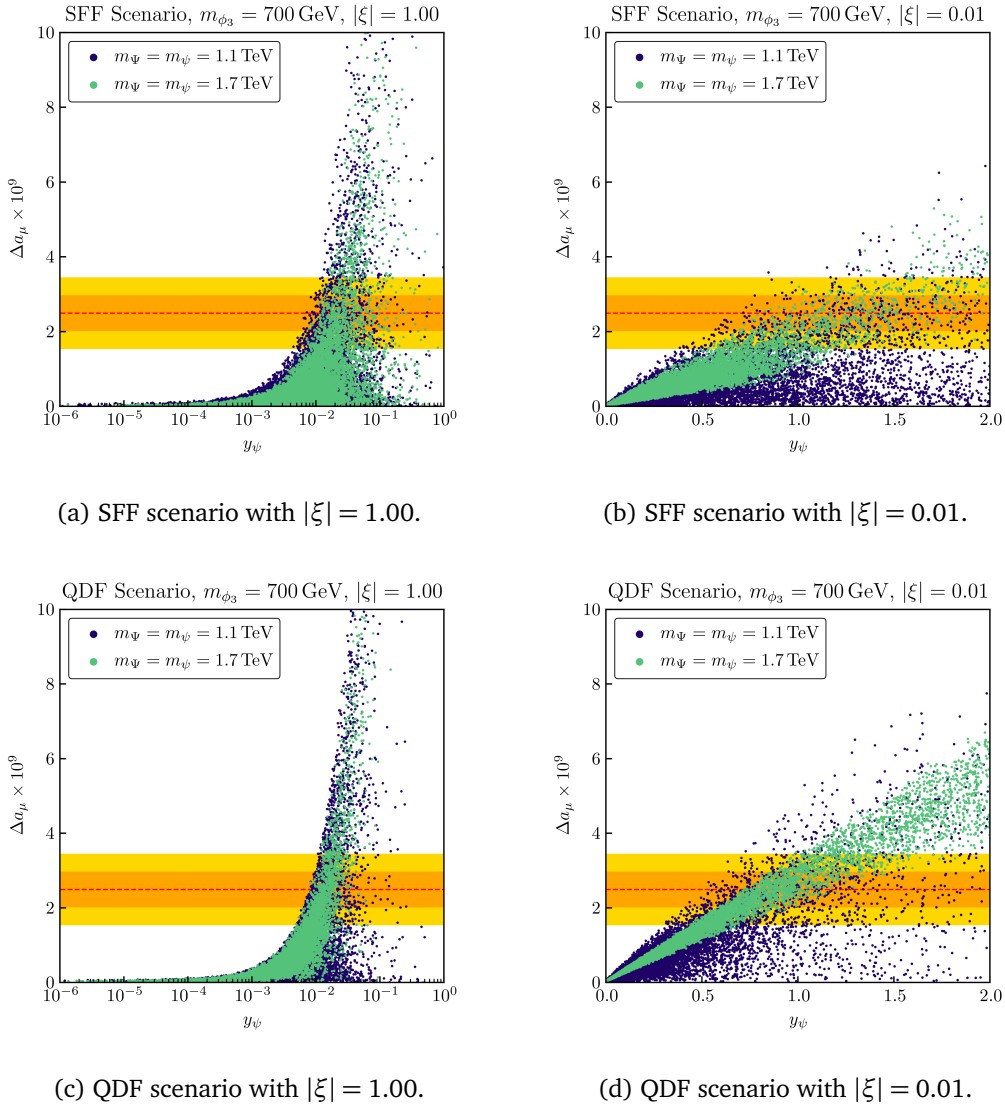

(a) SFF scenario with $|\xi| = 1.00$.

(b) SFF scenario with $|\xi| = 0.01$.

(c) QDF scenario with $|\xi| = 1.00$.

(d) QDF scenario with $|\xi| = 0.01$.

Figure 19: Dependence of the NP contributions $\Delta a_\mu$ on the Yukawa coupling $y_\psi$ for both freeze-out scenarios and both choices of $|\xi|$. The red dashed line shows the mean value of $\Delta a_\mu^{\rm exp}$ and the orange and yellow areas show the $1\sigma$ and $2\sigma$ bands, respectively.

the allowed points are slightly shifted to smaller values of $|\xi|$ for the QDF scenario, due to the slightly larger couplings allowed in this case. Increasing the value of $m_\Psi = m_\psi$ shrinks the area in the $|\xi| - y_\psi$ plane for which the experimental $2\sigma$ band can be reached. As large values $m_\Psi = m_\psi$ generally lead to larger couplings $|\lambda_{\mu i}|$ the upper edge of the area of allowed points is shifted towards smaller values of both $y_\psi$ and $|\xi|$, as growing couplings $|\lambda_{\mu i}|$ need either decreasing values of $y_\psi$ or $|\xi|$ in order to not yield a too large $\Delta a_\mu$. Since increasing mediator masses also suppress $\Delta a_\mu$, the lower edge is shifted towards larger values of $y_\psi$ and $|\xi|$.

The dependence of $\Delta a_\mu$ on the Yukawa coupling $y_\psi$ is shown for the SFF scenario in Figure 19a and Figure 19b. Here, the central value of $\Delta a_\mu^{\rm exp}$ can be reproduced for both choices of $|\xi|$. For $|\xi| = 0.01$ and masses $m_\psi = 1100\,{\rm GeV}$ this minimally requires $y_\psi \simeq 0.6$ while larger masses $m_\psi = 1700\,{\rm GeV}$ require $y_\psi \simeq 0.8$. If left-handed interactions are non-suppressed on the other hand, $y_\psi$ needs to be much smaller and we find $y_\psi \simeq 0.006$ for $m_\psi = 1100\,{\rm GeV}$ while one needs $y_\psi \simeq 0.008$ for $m_\psi = 1700\,{\rm GeV}$. This can be seen in Figure 19a. We further find

that large values of $m_\Psi = m_\psi$ shrink the area of possible values for $\Delta a_\mu$, while for $|\xi| = 0.01$ and a given value of $y_\psi$ it even leads to a lower limit on $\Delta a_\mu$. This limit arises because large mediator masses $m_\psi \gtrsim 1500\,\text{GeV}$ demand large DM–muon couplings $|\lambda_{\mu 3}| \gtrsim 1.0$, as can be seen in Figure 15b. Such a lower bound is not present for $|\xi| = 1.00$ as in this case even large mediator masses allow for small couplings $|\lambda_{\mu 3}|$, as can be seen in Figure 15a. Since growing mediator masses at the same time suppress $\Delta a_\mu$, the upper edge of accessible values shrinks for both cases of $|\xi|$ for increasing mediator masses.

We find the same behaviour also for the QDF scenario shown in Figure 19c and Figure 19d. Here, for $|\xi| = 0.01$ the central value of $\Delta a_\mu^{\text{exp}}$ can be reproduced for minimal values $y_\psi \simeq 0.5$ and masses $m_\psi = 1100\,\text{GeV}$, while we find $y_\psi \simeq 0.7$ for $m_\psi = 1700\,\text{GeV}$. In the non-suppressed case we find $y_\psi \simeq 0.008$ for $m_\psi = 1100\,\text{GeV}$ and $y_\psi \simeq 0.01$ for $m_\psi = 1700\,\text{GeV}$. We further find that for $|\xi| = 0.01$ and $m_\psi = 1700\,\text{GeV}$ the lower limit on $\Delta a_\mu$ is larger than for the SFF scenario, while the upper limit is smaller. The larger lower limit can be explained through the relic density constraint which requires all couplings $|\lambda_{\mu i}|$ to be large, in order to compensate for the flavour-averaging factor and not yield over-abundant dark matter. The reduced upper limit, on the other hand, is due to the fact that even for comparably small masses $m_\psi = 1100\,\text{GeV}$ the QDF scenario allows for close-to-maximal couplings $|\lambda_{\mu 3}| \simeq 1.7$. Hence, the increased suppression of $\Delta a_\mu$ for increased values of $m_\psi$ is less compensated by growing couplings $|\lambda_{\mu 3}|$ as they can maximally grow as large as $|\lambda_{\mu 3}| = 2.0$.

Finally, we also examine the correlation between NP contributions to $a_\mu$ and $m_\mu$ in Figure 20. To this end we show how large the corrections $\Delta a_\mu$ are for a given value of $|\Delta m_\mu|$ normalized to the physical muon mass $m_\mu$. We find that for both choices of $|\xi|$ the central value of $\Delta a_\mu^{\text{exp}}$ can be reached for corrections $|\Delta m_\mu|/m_\mu < 1$ in the SFF scenario, i.e. without introducing a fine-tuned muon mass. This also holds true for both choices of $m_\psi$ that are shown in Figure 20a and Figure 20b. We further find that non-suppressed left-handed interactions generally lead to larger corrections $|\Delta m_\mu|$ for sizeable NP effects in $a_\mu$. In this case most of the viable parameter points found by the combined analysis lie in the rage $y_\psi \sim \mathcal{O}(10^{-4}-10^{-1})$. As small values of $y_\psi$ increase the mass of the lightest charged mediator $m_{\psi_2}$, this suppresses both $\Delta a_\mu$ as well as $\Delta m_\mu$. However, since the function

$$\frac{m_{\psi_1}}{m_{\phi_k}^2} G(x_{k,1}) - \frac{m_{\psi_2}}{m_{\phi_k}^2} G(x_{k,2}) , \tag{90}$$

responsible for the suppression of $\Delta a_\mu$ is steeper than

$$m_{\psi_2} B_0(0, m_{\psi_2}, m_{\phi_k}) - m_{\psi_1} B_0(0, m_{\psi_1}, m_{\phi_k}) , \tag{91}$$

which causes the suppression of $|\Delta m_\mu|$, the slope of the distribution is steeper in Figure 20b than in Figure 20a. For the same reason we find an equivalent behaviour for increasing mediator masses, i.e. they again lead to larger contributions to $|\Delta m_\mu|$ for a given value of $\Delta a_\mu$.

The correlation between $\Delta a_\mu$ and $|\Delta m_\mu|$ is shown for the QDF scenario in Figure 20c and Figure 20d. Again, for both choices of $|\xi|$ the central value of $\Delta a_\mu^{\text{exp}}$ can be reached without exceeding the threshold $|\Delta m_\mu| > m_\mu$, i.e. without introducing a fine-tuned muon mass. While the correlation between $\Delta a_\mu$ and $|\Delta m_\mu|$ qualitatively shows the same behaviour as in the SFF scenario, we find that in the QDF scenario the allowed values lie on a thin band for the case of non-suppressed left-handed interactions shown in Figure 20c. This is due to the very small range of $y_\psi$ values that allow for sizeable $\Delta a_\mu$ in this case, as can be seen in Figure 19c. In that range the ratio $\Delta a_\mu/|\Delta m_\mu|$ is roughly constant in this freeze-out scenario, leading to the thin strips of Figure 20c. Note that we also checked if sizeable contributions to the electron or tau mass are generated and found those effects to be negligibly small. Hence, we conclude that in both scenarios and for both cases of $|\xi|$ our model is capable of accommodating $\Delta a_\mu^{\text{exp}}$ without introducing fine-tuned lepton masses and a fine-tuned muon mass in particular.

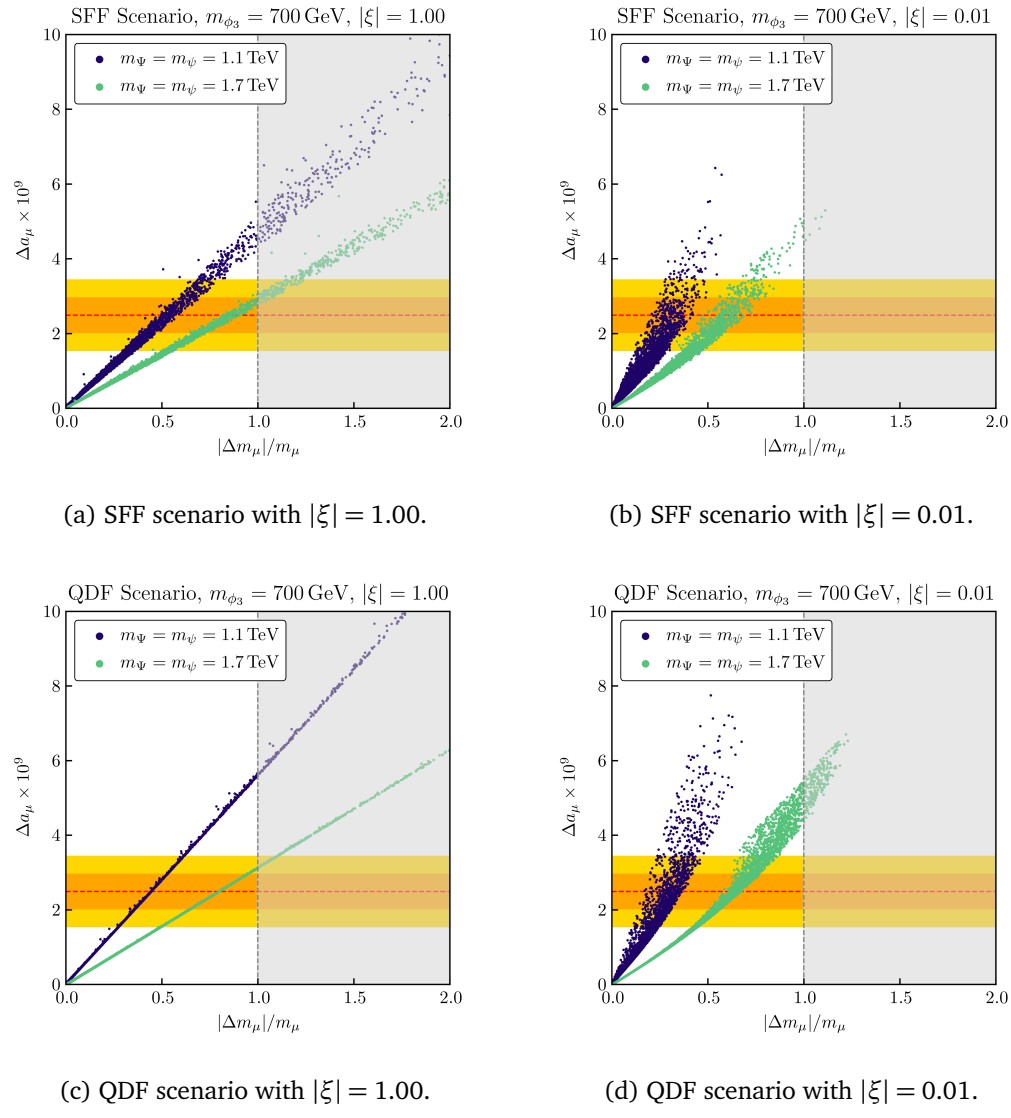

(a) SFF scenario with $|\xi| = 1.00$.

(b) SFF scenario with $|\xi| = 0.01$.

(c) QDF scenario with $|\xi| = 1.00$.

(d) QDF scenario with $|\xi| = 0.01$.

Figure 20: Correlation between $\Delta a_\mu$ and $|\Delta m_\mu|$ in both freeze-out scenarios and for both choices of $|\xi|$. The greyed-out area indicates the region with $|\Delta m_\mu|/m_\mu > 1$ which we consider fine-tuned. The red dashed line shows the mean value of $\Delta a_\mu^{\text{exp}}$ and the orange and yellow areas show the $1\sigma$ and $2\sigma$ bands, respectively.

## 10 Summary and Outlook

In this work we studied a simplified model of lepton-flavoured DM, in which a complex scalar DM flavour triplet couples to both right- and left-handed leptons. The interactions between DM and right-handed leptons are mediated by a vector-like charged Dirac fermion and governed by the $3 \times 3$ complex coupling matrix $\lambda$. Interactions between DM and left-handed leptons, on the other hand, are mediated by an $SU(2)_L$ doublet containing one charged and one neutral vector-like Dirac fermion. We assumed the coupling matrix of these interactions to be given by the same flavour-violating matrix $\lambda$ times a scaling parameter $\xi$. The presence of both left- and right-handed interactions between DM and the SM was found to lead to the absence of a dark flavour symmetry, implying that it does not fall into the DMFV class [20–26]. In turn, the DMFV connection between the coupling matrix $\lambda$ and the DM mass spectrum is lifted in

this model. Further, the two mediator fields interact with the SM Higgs doublet through the Yukawa coupling $y_\psi$.

To examine the structure of the coupling matrix $\lambda$, determine the model's viable parameter space and subsequently examine whether it is capable of generating sizeable effects in the muon anomalous magnetic moment $a_\mu$, we studied limits from collider searches, flavour experiments, precision tests of the SM, the DM relic density as well as DM detection experiments.

In Section 3 we explored constraints from LHC searches for sleptons in the same-flavour final state $\ell\bar{\ell} + \not{E}_T$ with $\ell = e, \mu$. We found that the interplay between contributions from processes with the two charged mediators leads to an exclusion of DM masses close to the equal mass threshold $m_{\phi_3} \approx m_{\psi_2}$ for values $0.5 \lesssim y_\psi \lesssim 1.00$, while this near-threshold exclusion tends to decrease with increasing values $y_\psi \gtrsim 1.00$. We obtained the largest exclusion in the $m_{\psi_2}-m_{\phi_3}$ plane for vanishing couplings $|\lambda_{\tau 3}|$ and maximum couplings to electrons and muons $|\lambda_{\ell 3}| = 2.0$. In this case the LHC constraints exclude mediator masses up to $m_{\psi_2} \simeq 750\,\text{GeV}$ and DM masses up to $m_{\phi_3} \simeq 400\,\text{GeV}$.

To determine the flavour structure of $\lambda$ we then studied limits from LFV decays in Section 4. Due to the contribution with a chirality flip in the loop governed by the Yukawa coupling $y_\psi$, the LFV decays $\ell_i \to \ell_j \gamma$ yield stringent bounds. The strongest limit is placed on the model by the decay $\mu \to e\gamma$ and we estimated the resulting constraint on the coupling matrix. Motivated by our goal to solve the $(g-2)_\mu$ anomaly, we satisfied this constraint by suppressing the DM–electron couplings and choosing $|\lambda_{ei}| \sim \mathcal{O}(10^{-6} - 10^{-1})$. Other LFV decays less severely constrain the coupling matrix $\lambda$ due to weaker experimental limits.

We continued to restrict the coupling matrix $\lambda$ in Section 5 by using constraints from precision tests of the SM. Here we particularly discussed the electron electric dipole moment $d_e$ as well as the electron magnetic dipole moment $a_e$, with NP contributions to the latter having the same sign in our model as the NP effects in $a_\mu$. We found that once the DM–electron couplings are suppressed according to the LFV constraints, the EDM constraint allows for $\mathcal{O}(1)$ imaginary parts of the scaling parameter $\xi$. Moreover, for accordingly suppressed DM–electron couplings the constraints on the electron MDM $a_e$ are automatically fulfilled.

In Section 6 we studied in which part of our model's parameter space the observed DM relic density can be reproduced. We investigated two benchmark scenarios for the thermal freeze-out: the QDF scenario in which negligible mass splittings lead to the presence of all dark flavours during freeze-out, and the SFF scenario where due to a significant mass splitting between the DM flavours only the lightest state contributes to the freeze-out. For both scenarios, we studied the two cases of suppressed and non-suppressed left-handed interactions between DM and the SM with $|\xi| = 0.01$ and $|\xi| = 1.00$, respectively. For small left-handed couplings the DM annihilation rate is $p$-wave suppressed while for sizeable left-handed interactions this suppression is lifted. Still the relic density constraint allows for large couplings $|\lambda_{ij}| \sim \mathcal{O}(1)$ in both freeze-out scenarios and both cases for $\xi$.

The direct detection phenomenology of our model was studied in Section 7.1, where we used XENON1T data to constrain the coupling matrix $\lambda$. As the photon penguin that dominates DM–nucleon scattering is proportional to the logarithm of the mass of the lepton in the loop, the restrictions on $\lambda$ were found to increase for decreasing lepton masses. The largest constraints were thus found for the DM–muon coupling, since the DM–electron coupling is already suppressed due to the flavour constraints.

Section 7.2 was dedicated to the limits from indirect detection experiments. Here we focussed on limits from AMS measurements of the positron flux as well as measurements of the $\gamma$-ray line and continuum spectrum performed by the Fermi-LAT satellite and the H.E.S.S. telescope. For non-suppressed left-handed interactions the constraints from measurements of the positron flux and the $\gamma$-ray continuum spectrum are restrictive for DM masses $m_{\phi_3} \lesssim 500\,\text{GeV}$

and $m_{\phi_3} \lesssim 700\,\text{GeV}$, respectively. The $\gamma$-ray line spectrum generally leads to relevant constraints in the near-degeneracy region $m_{\phi_3} \approx m_{\psi_2}$. In total, constraints from indirect detection were still found to be weak in comparison to other limits.

To obtain our model's viable parameter space we then performed a combined analysis in Section 8 by demanding that all constraints are simultaneously fulfilled at the $2\sigma$ level. The global picture is mainly determined by flavour, relic density and direct detection constraints. For the SFF scenario the combination of flavour and relic density constraints forces the allowed points to lie in a band in the $|\lambda_{\tau 3}|-|\lambda_{\mu 3}|$ plane, while the direct detection constraint dominates in a small part of the parameter space over the relic density constraint. In these regions, the outer edge of the mentioned bands shrinks toward the inner band for growing values of $|\lambda_{\mu 3}|$. For the QDF scenario we found that for large parts of the parameter space the direct detection constraint dominates over the relic density constraint. Consequently the combined analysis also allows for simultaneously small values of $|\lambda_{\tau 3}|$ and $|\lambda_{\mu 3}|$, since the relic density constraint in this case can also be fulfilled through annihilations of the heavier states $\phi_1$ and $\phi_2$ alone, provided that the mediator mass is sufficiently small, $m_{\psi_2} \lesssim 1500\,\text{GeV}$. Generally the case of non-suppressed left-handed interactions with $|\xi| = 1.00$ allows for smaller couplings $|\lambda_{\tau 3}|$ and $|\lambda_{\mu 3}|$ than the case with $|\xi| = 0.01$.

Finally, we used our results from Section 8 to examine in Section 9 if our model is able to account for the discrepancy between the SM and experiment in the muon anomalous magnetic moment $a_\mu$. To this end we calculated $\Delta a_\mu$ in the regions identified as viable in the combined analysis and compared it with the experimental value. We further evaluated accompanying corrections $\Delta m_\mu$ to the muon mass and checked if sizeable effects in $a_\mu$ introduce a fine-tuned muon mass. We found that in both freeze-out scenarios the central value of $\Delta a_\mu^{\text{exp}}$ can be reached within the region of parameter space that we regard as non-fine-tuned for both cases of $\xi$, requiring different values for the mediator-Higgs coupling $y_\psi$. Noteworthy, for non-suppressed left-handed interactions larger corrections to the muon mass are generated for a given value of $\Delta a_\mu$ than for $|\xi| = 0.01$.

We conclude that lepton-flavoured DM with couplings to both left- and right-handed leptons accompanied by Higgs portal interactions of the corresponding mediators elegantly connects the current most convincing hints at NP: the DM problem and the muon $(g-2)$ anomaly. In spite of exhibiting a very rich phenomenology spanning over several branches of particle physics and thus being subject to many constraints, this model still allows for a joint explanation of both. Hence, it qualifies as an attractive DM candidate waiting to be further probed with increased sensitivity by future experiments.

## Acknowledgments

We thank Manuel Egner, Mustafa Tabet and Marco Fedele for useful discussions.

**Funding information** This work is supported by the Deutsche Forschungsgemeinschaft (DFG, German Research Foundation) under grant 396021762 – TRR 257. P.A. is supported by the STFC under Grant No. ST/T000864/1. H.A. acknowledges the scholarship and support he receives from the Avicenna-Studienwerk e.V., the support of the doctoral school "Karlsruhe School of Elementary and Astroparticle Physics: Science and Technology (KSETA)" and the funding he received for his academic visit at the University of Oxford from the "Karlsruhe House of Young Scientists (KHYS)". He further acknowledges the hospitality during his academic visit at the Rudolph Peierls Centre for Theoretical Physics.

## A    Relic density

The functions $A_{ijkl}, B_{ijkl}, C_{kl}$ and $D_{kl}$ from eqs. (49) –(52) read

$$A_{ijkl} = (m_{\phi_j}^2 - m_{\ell_l}^2 - t)(t + m_{\ell_k}^2 - m_{\phi_i}^2) - t(s - m_{\ell_k}^2 - + m_{\ell_l}^2), \tag{A.1}$$

$$B_{ijkl} = \xi^* m_{\ell_l}(m_{\phi_i}^2 - m_{\ell_k}^2 - t) + \xi m_{\ell_k}(m_{\phi_j}^2 - m_{\ell_l}^2 - t), \tag{A.2}$$

$$C_{kl} = -2m_{\ell_k} m_{\ell_l} t, \tag{A.3}$$

$$D_{kl} = 2|\xi|^2(s - m_{\ell_k}^2 - m_{\ell_l}^2) - 2m_{\ell_k} m_{\ell_l}\left(\xi^{*2} + \xi^2\right). \tag{A.4}$$

The $p$-wave contribution to the thermally averaged annihilation cross section from eq. (53) for $\xi \neq 0$ is given by

$$
\begin{aligned}
b = \frac{1}{9}\sum_{ijkl}\frac{|\lambda_{ik}|^2|\lambda_{jl}|^2}{32\pi m_{\phi_3}^2}\Bigg\{ & \frac{\left(2 + \mu_2^2 + \mu_1^2 + \left(\mu_1^2 - \mu_2^2\right)\cos 2\theta_\psi\right)^2}{\left(1 + \mu_2^2\right)^2\left(1 + \mu_1^2\right)^2} \\
& - \frac{|\xi|^2\sin^2 2\theta_\psi(\mu_2 - \mu_1)^2}{(1 + \mu_2^2)^4(1 + \mu_1^2)^4}\Big[3\left(\mu_2^6 + 6\mu_2^4 + \mu_1^2\right)\mu_2^6 \\
& + 2\mu_1\left(\mu_1^2 - 5\right)\left(3\mu_1^2 + 1\right)\mu_2^5 + \left(18\mu_1^6 + 31\mu_1^4 + 4\mu_1^2 + 3\right)\mu_2^4 \\
& - 4\mu_1\left(7\mu_1^4 + 26\mu_1^2 + 7\right)\mu_2^3 + \left(3\mu_1^6 + 4\mu_1^4 + 31\mu_1^2 + 18\right)\mu_2^2 \\
& - 2\mu_1\left(\mu_1^2 + 3\right)\left(5\mu_1^2 - 1\right)\mu_2 + 3\mu_1^2\left(\mu_1^2 + 6\right) + 3\Big] \\
& + 4|\xi|^4\Bigg[\frac{\cos^4\left(\theta_\psi\right)}{\left(\mu_1^2 + 1\right)^2} + \frac{\sin^2\left(\theta_\psi\right)}{\left(\mu_2^2 + 1\right)^2}\Bigg(\frac{2\left(\mu_2^2 + 1\right)\cos^2\left(\theta_\psi\right)}{\mu_1^2 + 1} \\
& + \sin^2\left(\theta_\psi\right)\Bigg) + \frac{1}{\left(\mu_0^2 + 1\right)^2}\Bigg]\Bigg\},
\end{aligned}
\tag{A.5}
$$

in the limit of equal initial state masses and vanishing final state masses. Here we have used $\mu_\alpha = m_{\psi_\alpha}/m_{\phi_3}$.

## B    Direct detection

The DM–nucleon cross section for $t$-channel scatterings through the Higgs portal reads [81]

$$\sigma_{\text{SI}}^{N,H\phi} = \frac{\lambda_{H\phi}^2 y_N^2}{4\pi}\frac{\mu^2 m_N^2}{m_H^4 m_{\phi_3}^2}, \tag{B.1}$$

where $m_N$ is the nucleon mass and $\mu = m_{\phi_3} m_N/(m_{\phi_3} + m_N)$ is the reduced mass of the DM–nucleon system. In order to estimate in which parts of the parameter space these contributions grow larger than the photon one-loop penguin from Figure 11a that we consider in the numerical analysis, we set the couplings to $|\lambda_{i3}| = 2$ and $|\xi| = 1$ as well as $y_\psi = 0$ in eq. (64) for maximum mixing with $\theta_\psi = \pi/4$. Comparing with eq. (B.1) then gives the maximum allowed value of the Higgs portal coupling $\lambda_{H\phi}$ for at most equal scattering cross sections. This illustrated by the contours shown in Figure 21.

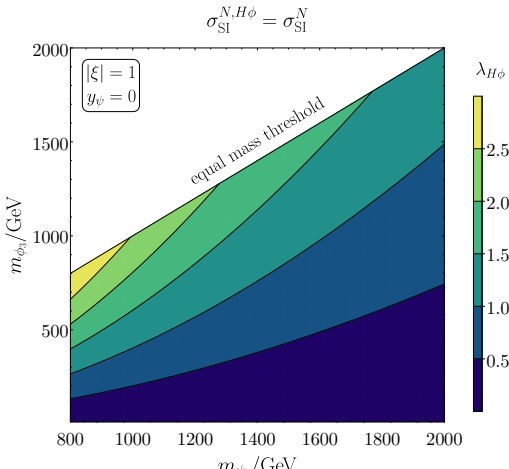

**Figure 21:** Maximum allowed values of $\lambda_{H\phi}$ in order to give at most equal contributions to the DM–nucleon scattering cross section as the photon penguin diagram.

## C  Indirect detection

The expression for the interference term of $\psi_1$ and $\psi_2$ for the internal bremsstrahlung process of Figure 12a is given by

$$
\begin{aligned}
\langle \sigma v \rangle_{\ell\bar{\ell}\gamma}^{12} = \sum_{ij} &\frac{-\alpha_{\mathrm{em}}|\lambda_{i3}|^2|\lambda_{j3}|^2\left(c_\theta^2 + |\xi|^2 s_\theta^2\right)\left(s_\theta^2 + |\xi|^2 c_\theta^2\right)}{128 m_{\phi_3}^2 \pi^2(\mu_2 - \mu_1)^2}\bigg\{ -4p_1\mu_2(1+\mu_2)^2 \\
&+ 4p_2\mu_2(1+\mu_2)^2 - 2(\mu_2 - \mu_1)^2 - 4p_3\mu_1(1+\mu_1)^2 + 4p_4\mu_1(1+\mu_1)^2 \\
&+ p_5(\mu_2 + \mu_1)(2 + \mu_2(2+\mu_2) + \mu_1(2+\mu_1)) \\
&- p_6(\mu_2 + \mu_1)(2 + \mu_2(2+\mu_2) + \mu_1(2+\mu_1)) \\
&+ p_7(\mu_2 + \mu_1)(2 + \mu_2(2+\mu_2) + \mu_1(2+\mu_1)) \\
&- p_8(\mu_2 + \mu_1)(2 + \mu_2(2+\mu_2) + \mu_1(2+\mu_1)) + \mu_2^3(4l_1^2 - l_2l_3 + l_3l_4 - 3l_5 - l_3l_5 \\
&+ l_4l_5 + 4l_5l_6 + l_1(3 + l_2 + l_3 - 2l_4 - 4l_5 - 4l_6 - l_8) + l_5l_8) + \mu_2^2(8l_1^2 - 2l_2(1+l_3) \\
&+ 2l_4(1 + l_3 + l_5) - 2l_5(l_3 - 4l_6) + 2l_1(l_2 + l_3 - 2(l_4 + 2(l_5 + l_6))) - l_1l_9 + l_5l_9 \\
&+ (-l_2(-2+l_3) + l_1(-1 + l_2 + l_3 - 2l_4) + (-2 + l_3)l_4 + (1 - l_3 + l_4)l_5)\mu_1) \\
&+ \mu_2(4l_1^2 - l_2(3 + 2l_3) + 2l_3l_4 - 2l_3l_5 + 2l_4l_5 + 3(l_4 + l_5) + 4l_5l_6 \\
&+ l_1(-3 + 2l_2 + 2l_3 - 4l_4 - 4l_5 - 4l_6 - l_8) + l_5l_8 + \mu_1(2(l_2 - 2l_2l_3 \\
&+ l_1(-1 + 2l_2 + 2l_3 - 4l_4) - l_4 + 2l_3l_4 + l_5 - 2l_3l_5 + 2l_4l_5) \\
&+ (l_2 - l_2l_3 + l_1(-2 + l_2 + l_3 - 2l_4) + (-1 + l_3)l_4 + (2 - l_3 + l_4)l_5)\mu_1)) \\
&+ \mu_1(l_1(3 + 2l_2 + 2l_3 - 4l_4) + 2l_3l_4 + 4l_4^2 - 2l_3l_5 + 2l_4l_5 - 3(l_4 + l_5) - 4l_4l_7 - l_4l_8 \\
&+ l_2(3 - 2l_3 - 4l_4 + 4l_7 + l_8) + \mu_1(2l_1(1 + l_2 + l_3 - 2l_4) \\
&- 2(l_5 + l_3(-l_4 + l_5) + l_2(l_3 + 4l_4 - 4l_7) - l_4(4l_4 + l_5 - 4l_7)) + (l_2 - l_4)l_9 \\
&+ (l_1(l_3 - 2l_4) + 3l_4 + l_3l_4 + 4l_4^2 - l_3l_5 + l_4l_5 - 4l_4l_7 - l_4l_8 \\
&+ l_2(-3 + l_1 - l_3 - 4l_4 + 4l_7 + l_8))\mu_1))\bigg\},
\end{aligned}
\tag{C.1}
$$

with the logarithms $l_i$ and polylogarithms $p_i$ defined as

$$
l_1 = \log(1 + \mu_2), \quad l_2 = \log(\mu_1 - 1), \quad l_3 = \log(\mu_2 + \mu_1),
$$

$$l_4 = \log(1 + \mu_1), \quad l_5 = \log(\mu_2 - 1), \quad l_6 = \log(\mu_2),$$
$$l_7 = \log(\mu_1), \qquad l_8 = \log(16), \qquad l_9 = \log(256), \tag{C.2}$$

and

$$p_1 = \text{Li}_2\left(\frac{\mu_2 - 1}{2\mu_2}\right), \quad p_2 = \text{Li}_2\left(\frac{1 + \mu_2}{2\mu_2}\right), \quad p_3 = \text{Li}_2\left(\frac{\mu_1 - 1}{2\mu_1}\right),$$
$$p_4 = \text{Li}_2\left(\frac{1 + \mu_1}{2\mu_1}\right), \quad p_5 = \text{Li}_2\left(\frac{\mu_2 - 1}{\mu_2 + \mu_1}\right), \quad p_6 = \text{Li}_2\left(\frac{1 + \mu_2}{\mu_2 + \mu_1}\right),$$
$$p_7 = \text{Li}_2\left(\frac{\mu_1 - 1}{\mu_2 + \mu_1}\right), \quad p_8 = \text{Li}_2\left(\frac{1 + \mu_1}{\mu_2 + \mu_1}\right). \tag{C.3}$$

The constraint coming from indirect detection experiments for the case of suppressed left-handed interactions is shown in Figure 22.

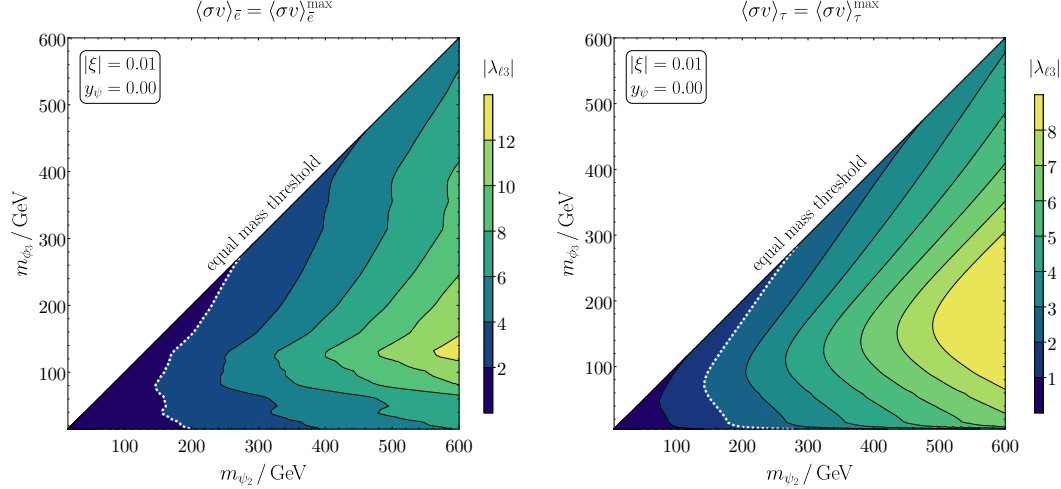

(a) constraints from the positron flux.

(b) constraints from the $\gamma$ continuum spectrum.

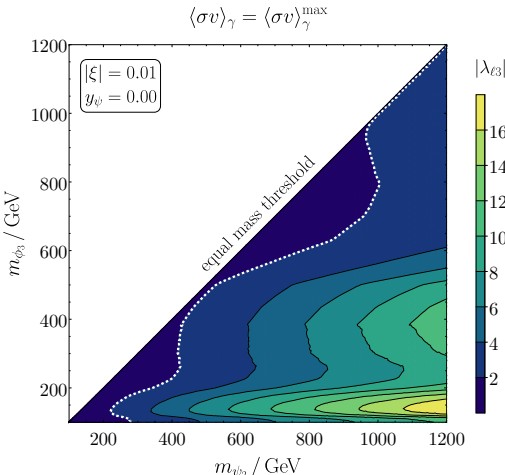

(c) constraints from the $\gamma$ line spectrum.

Figure 22: Restrictions on the model parameters from indirect detection experiments for suppressed left-handed interactions. In all three plots we have assumed maximum mixing with $\theta_\psi = \pi/4$. The area included by the white dashed line and the equal mass diagonal indicates in which mass regime the constraints are relevant.

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
