# Peer review of "Flavoured $(g-2)_μ$ with Dark Lepton Seasoning"

_SciPost Physics, doi:SciPost Phys. 15, 176 (2023)_

## Round 2 · Referee Report · Anonymous (Referee 2) · 2023-8-28

Report

In this manuscript, the authors perform a comprehensive analysis of a specific model with new scalars and heavy leptons that simultaneously explains the current observation of the muon (g-2) and provides for a viable dark matter candidate. The work is an extension of an earlier project; the difference is an extension of the particle content and less assumptions on the flavor structure of the introduced couplings.

The paper addresses two important open questions in particle physics and is thus in general of relevance. However, the model has 26 parameters, so that many assumptions need to be made to make a phenomenological statement. Moreover, the considered observables and their interpretation are standard and do not suggest new search directions. In this sense, I find the information gain compared to previous work rather modest.

The presentation in the manuscript is very clear and the analysis of the various observables is conducted in a solid, careful way. I would like to invite the authors to consider the following questions and comments:

1) The Lagrangian in (2.1) assumes, as far as I can see, a Z2 symmetry in the dark sector, which is motivated later in the text with the stability of the DM candidate. It would be good to make this statement already around (2.1); otherwise the Lagrangian would allow for additional interaction terms.

2) What is the motivation for assuming the hierarchy in (2.12)?

3) Among the collider observables considered in Sec. 3, I would expect that effects of virtual dark sector particles on Higgs and electroweak observables play a significant role in setting bounds on the model parameters. See e.g. https://arxiv.org/abs/1810.10993 or https://arxiv.org/abs/1708.01614. Later in the text, the authors mention that electroweak precision observables should be negligible in this respect. I would appreciate a quantitative discussion of this point.

4) For heavy leptons with couplings to both left- and right-handed SM leptons, I would expect loop corrections to the SM lepton masses that do not scale with the bare mass. With a general flavor structure, these contributions could be undesirably large and require fine-tuning with the bare mass to obtain the observed masses. Also here, I would appreciate a discussion of this issue.

5) To determine the relic abundance from dark matter freeze-out, the authors consider two specific scenarios. What is the motivation for precisely these benchmarks?

6) In the QDF scenario, the dark scalars are nearly degenerate in mass. Are loop contributions to the mass splitting relevant in the parameter space that produces the right relic abundance? Since the (co-)annihilation rate is exponentially sensitive to the mass splitting if several annihilation processes contribute to setting the relic abundance, I would expect that even a small change in the mass splitting could have a remarkable effect.

---

## Round 2 · Referee Report · Anonymous (Referee 4) · 2023-9-5

Report

The authors of manuscript 2212.08142 explore a model extending the Standard Model by scalar and vector-like fermion to address the dark matter and the muon $(g-2)$ problem within a unified framework. The study involves a detailed investigation, restricted by choice of parameter space, of phenomenology while investigating constraints from collider searches, flavor experiments, indirect and direct detection analysis, etc. For the paper to be suitable for publication, the authors should consider incorporating the following points.

  1. Update the reference with new results for muon $g-2$.

  2. The model consists of three copies of scalar fields, a vector-like doublet, and a singlet. With these new fields, there should be more terms in the Lagrangian in Eq.~(2.1); it should be clearly stated how these terms are forbidden in Sec.~(2.1). Moreover, the authors consider three complex scalar fields; would it not be minimal to just one scalar field?

  3. In Eq.(2.11), the mass for the scalar fields should have additional term proportional $\lambda_{H\phi}$.

  4. Numerous assumptions are made regarding the choice of the masses. How does the analysis vary for different choices? For instance, does the model allow fermionic DM if $m_{\psi_0}$ becomes the lightest particle? If so, the authors should briefly comment on how this can be accomplished without delving into a detailed study.

  5. On page 8, in the first paragraph of sec~3.1, line 8, the authors mention that $3\ell$+MET is strongly constrained due to lepton flavor violation (LFV). Expanding this discussion by including specific examples on LFV observables would be good.

  6. In sec.~(3.2), how are signal cross-sections being compared with the experimental upper limit? More detail on signal and background estimates would further enhance this section. In addition, in Fig.~(3.3) (d), why does the limit around $m_{\psi_2} = 400$ GeV gets weaker?

  7. In sec.~(4.2), the authors have tuned the nonchiral part of LFV to zero without any proper explanation.

  8. In equation.~(6.3) the coannihilation channels does not suffers much suppression when the $m_\psi$ is close to $m_\phi$. It would be good to comment on this and mention why these corners of the parameter are not included. Moreover, below Eq.(6.3), the authors argued that DM annihilation via SM Higgs can be neglected. However, the authors state that they have chosen $\lambda_{H\phi}$ such that the tree-level and one-loop contribution cancel in Direct detection. If so, why is DM annihilation via $\lambda_{H\phi}$ not considered?

  9. DM with large Yukawa couplings with $Z_2$ odd fermion can also drive its bare mass squared term to negative values via RGE, potentially breaking the discrete symmetry responsible for stabilizing DM. Authors should check if their parameter space leads to such issues or not.

  10. On page 37 sec~(9.2), the authors mention ``one needs small Yukawa couplings $y_\psi \lesssim 10^{-1}$ in order to stay within the $2\sigma$ band ..". However, it is unclear how small this coupling can be. Perhaps the plot would have been more clearer if the $y_\psi$ was in log scale.

  11. Incorporating $(g-2)_\mu$ within the model while maintaining perturbative couplings should give an upper limit on the new physics scale. What is the range of the masses and couplings that one can have in this model while incorporating DM and $(g-2)_\mu$? Comments on this matter greatly enhance the paper.

---

## Round 3 · Referee Report · Anonymous (Referee 2) · 2023-9-27

Report

I thank the authors for answering my questions and for making adjustments in the manuscript. Besides point 6), all of my points have been addressed to my satisfaction.

Regarding point 6), the authors reply that the mass parameters denote the renormalized masses. I find this assignment problematic, because the mass *differences* are physical and not a matter of renormalization scheme choice. Setting the Lagrangian mass parameters equal to the renormalized particle masses makes it impossible to interpret the numerical results in terms of the model parameters. I invite the authors to reflect on this issue before publication.

---

## Round 3 · Referee Report · Anonymous (Referee 4) · 2023-10-4

Report

The authors have effectively addressed all the concerns raised, incorporating changes to both the text and figures. Consequently, I recommend this manuscript for publication.

---

## Round 3 · Author Response

\noindent Dear Editor,

\vskip 0.3cm

We thank the referees for reading our manuscript and for making valuable suggestions for improvement. Below please find our reply to the comments. We hope that with these modifications, our paper can be published in SciPost.\vspace{3mm}

{\bf Comments from Referee 1:}
\begin{enumerate}
\item {\it The Lagrangian in (2.1) assumes, as far as I can see, a Z2 symmetry in the dark sector, which is motivated later in the text with the stability of the DM candidate. It would be good to make this statement already around (2.1); otherwise the Lagrangian would allow for additional interaction terms.}

We thank the referee for this suggestion and have added a sentence just above (2.1).

\item {\it What is the motivation for assuming the hierarchy in (2.12)?}

Assuming $m_{\phi_3}$ to be the lightest NP state is necessary to ensure its stability. Other than that, the only assumption that enters (2.12) is a naming convention. The mixing of the charged mediators results in one of them, to be named $\psi_1$, to be heavier than the neutral mediator $\psi_0$, and the other one, named $\psi_2$, to be lighter. We have rephrased the text above (2.12) to clarify this.

\item {\it Among the collider observables considered in Sec. 3, I would expect that effects of virtual dark sector particles on Higgs and electroweak observables play a significant role in setting bounds on the model parameters. See e.g. https://arxiv.org/abs/1810.10993 or https://arxiv.org/abs/1708.01614. Later in the text, the authors mention that electroweak precision observables should be negligible in this respect. I would appreciate a quantitative discussion of this point.}

We thank the referee for this suggestion. We have now provided a more detailed discussion of Higgs and electroweak precision observables in Section 5.3 indicating why the currently available constraints are not competitive with other bounds on the parameter space of our model. A discussion of these observables in the context of future colliders as provided in https://arxiv.org/abs/1810.10993 or https://arxiv.org/abs/1708.01614, while interesting, is beyond the scope of the present paper.

\item {\it For heavy leptons with couplings to both left- and right-handed SM leptons, I would expect loop corrections to the SM lepton masses that do not scale with the bare mass. With a general flavor structure, these contributions could be undesirably large and require fine-tuning with the bare mass to obtain the observed masses. Also here, I would appreciate a discussion of this issue.}

The referee is correct that loop corrections to the SM lepton masses arise in our model that do not scale with the bare lepton masses and hence potentially pose a fine-tuning problem. This issue is particularly relevant in conjunction with addressing the $(g-2)_\mu$ anomaly, since the latter requires large muon couplings. Electron couplings, on the other hand, were found small in our analysis. Tau couplings can be sizeable, however due to the larger bare tau mass the issue is less relevant.
The potential fine-tuning issue in the muon mass was highlighted in Section 9, together with the NP contribution to the muon anomalous magnetic moment. Specifically, this is discussed in Equation (9.12) and Figure 9.4 and the accompanying text.

\item {\it To determine the relic abundance from dark matter freeze-out, the authors consider two specific scenarios. What is the motivation for precisely these benchmarks?}

The two freeze-out scenarios -- quasi-degenerate freeze-out (QDF) and single flavour freeze-out (SFF) -- are two limiting benchmark cases for the dynamics of the DM thermal freeze-out. These scenarios have been used as benchmarks in the previous studies of flavoured DM beyond MFV, see Refs. [20-26]. While considerably simplifying the dynamics of flavoured DM freeze-out, these scenarios allow to capture the main phenomenological features of the model. Sticking to these two benchmark models also allows for a direct comparison of the results with the ones of Reference [26], where lepton-flavoured scalar DM coupling exclusively to right-handed leptons has been studied.
We note that in general the freeze-out dynamics of flavoured DM can be rather involved and is the subject of a separate ongoing study [69].
We added an explanation on page 18 to better motivate these two scenarios.

\item {\it In the QDF scenario, the dark scalars are nearly degenerate in mass. Are loop contributions to the mass splitting relevant in the parameter space that produces the right relic abundance? Since the (co-)annihilation rate is exponentially sensitive to the mass splitting if several annihilation processes contribute to setting the relic abundance, I would expect that even a small change in the mass splitting could have a remarkable effect.}

Loop corrections to the dark scalar masses may indeed affect their splitting and thus have an impact on whether a given parameter point indeed falls into one of the benchmark freeze-out scenarios considered by us -- QDF for mass splittings below 1\% or SFF for mass splittings greater than 10\% -- and the respective approximations hold. In order to avoid ambiguities, we therefore take the mass parameters $m_{\phi_1}$ to be the renormalised on-shell masses. We have added a footnote on page 18.
\end{enumerate}\vspace{2mm}

{\bf Comments from Referee 2:}
\begin{enumerate}
\item {\it Update the reference with new results for muon $(g-2)$.}

While our paper had been submitted to SciPost well before the announcement of this new result, we have decided to follow the referee's suggestion and include the new measurement in our analysis. Figures 9.2-9.4 have been updated, and the text has been modified accordingly.
\item {\it The model consists of three copies of scalar fields, a vector-like doublet, and a singlet. With these new fields, there should be more terms in the Lagrangian in Eq. (2.1); it should be clearly stated how these terms are forbidden in Sec.~(2.1).}

The model assumes a $\mathbb{Z}_2$ symmetry, under which the new particles are charged, which forbids additional terms in the Lagrangian. We have added a sentence above (2.1) for clarification.

{\it Moreover, the authors consider three complex scalar fields; would it not be minimal to just one scalar field?}

The goal of our work is to investigate a model of {\bf flavoured} DM with couplings to both left- and right handed leptons and its implications for phenomenology, as stated clearly in the abstract and introduction. The introduction of DM as a flavour triplet is therefore minimal within the scope of our work.

\item {\it In Eq.~(2.11), the mass for the scalar fields should have additional term proportional $\lambda_{H\phi}$.}

Indeed in the most general case this term should be present. We omitted it due to our choice to neglect the Higgs portal coupling $\lambda_{H\phi}$ in our analysis, i.\,e.\ we assume its renormalised value to be small. We added a footnote for clarification.

\item {\it Numerous assumptions are made regarding the choice of the masses. How does the analysis vary for different choices? For instance, does the model allow fermionic DM if $m_{\psi_0}$ becomes the lightest particle? If so, the authors should briefly comment on how this can be accomplished without delving into a detailed study.}

In fact, the only assumption is that the lightest dark scalar is the lightest new particle in the spectrum. The mass splitting due to the mixing of the charged mediators always results in one charged mediator, $\psi_2$ in our convention, to be lighter than the neutral mediator $\psi_0$. It is thus not possible to have $\psi_0$ as a DM candidate. As a side remark, we note that DM as the neutral component of an EW doublet would anyway be excluded by direct detection experiments.

We have added a clarification above eq. (2.12).

\item {\it On page 8, in the first paragraph of sec~3.1, line 8, the authors mention that $3\ell+$MET is strongly constrained due to lepton flavor violation (LFV). Expanding this discussion by including specific examples on LFV observables would be good.}

We meant to say here that for the $3\ell+$MET signatures with three different lepton flavours, i,\,e.\ an electron, a muon and a tau, no dedicated LHC searches are available, since in many models, in particular supersymmetry, such final states are correlated with the strongly constrained LFV decays. We have rephrased the corresponding statement in the paper.

\item {\it In sec.~(3.2), how are signal cross-sections being compared with the experimental upper limit? More detail on signal and background estimates would further enhance this section.}

Since the CMS search in [39], using the full run 2 data set, places the strongest constraints on the parameter space of our model, we have recasted this search. To compare the signal cross-section predicted in our model with the experimental bound, we obtained the latter from the SModelS database. Performing our own analysis with cuts and backgroud estimates is therefore not necessary.
On a practical level we implemented the Lagrangian from eq. (2.1) in FeynRules.
Using this implementation we generated a UFO file [45] and calculated the leading-order signal
cross section of the relevant process in MadGraph 5. To constrain the parameter space of
our model we then compared the signal cross section to the experimental upper limit obtained
from the above mentioned search. In doing this, we neglected the impact of the potentially
different final-state kinematics due to the different spin-statistics in our model relative to the
SUSY case.

We believe that this procedure is clearly detailed in the first two paragraphs of section 3.2.

{\it In addition, in Fig.~(3.3) (d), why does the limit around $m_{\psi_2} = 400$\,GeV gets weaker?}

The exclusion in the soft region is obtained from pair production and decay of the heavier mediator, and its precise shape depends on the mass splitting between the two charged mediators. Practically, the exclusion contour stemming from processes involving the heavy mediator moves to the left with increasing splitting, giving rise to the change in shape.

We believe that this feature is clearly described in the discussion of Figure 3.3, see page 9 and 10.

\item {\it In sec.~(4.2), the authors have tuned the nonchiral part of LFV to zero without any proper explanation.}

We have omitted the subleading chirality-preserving contribution only in the semi-analytical estimate in eqs. (4.13), (4.14). Our numerical analysis includes the full expressions from section 4.1. We have added a sentence below (4.14) to stress this.

\item {\it In equation (6.3) the coannihilation channels does not suffers much suppression when the $m_\psi$ is close to $m_\phi$. It would be good to comment on this and mention why these corners of the parameter are not included.}

Due to the exponential nature of the Boltzmann suppression, these channels are irrelevant outside of the highly fine-tuned parameter region $m_{\phi_3}\simeq m_{\psi_2}$ which we omit in our analysis. We have added a sentence below eq.\ (6.3).

{\it Moreover, below Eq. (6.3), the authors argued that DM annihilation via SM Higgs can be neglected. However, the authors state that they have chosen $\lambda_{H\phi}$, such that the tree-level and one-loop contribution cancel in Direct detection. If so, why is DM annihilation via $\lambda_{H\phi}$ not considered?}

To be more precise, we have chosen the renormalised coupling $\lambda_{H\phi}$ to be small, such that its effects can be neglected. We have clarified this below eq. (6.3).

\item {\it DM with large Yukawa couplings with $\mathbb{Z}_2$ odd fermion can also drive its bare mass squared term to negative
values via RGE, potentially breaking the discrete symmetry responsible for stabilizing DM. Authors should check if their parameter space leads to such issues or not.}

The scalar dark matter gets quantum corrections to its mass both from running contributions from loops of fermions, as the referee suggests, as well as from UV thresholds, which are incalculable within the EFT and are at the root of fine-tuning problems for light scalars. Depending upon the relative size of the bare mass term and these contributions the $\mathbb{Z}_2$ symmetry can generically remain unbroken. We treat our scalar masses $m_{\phi_i}$ as the renormalised on-shell masses to avoid ambiguities. We have added a footnote on page 6 to clarify our choice.

\item {\it On page 37, sec~(9.2), the authors mention one needs small Yukawa couplings $y_\psi < 10^{-1}$ in order to stay within the $2\sigma$ band..". However, it is unclear how small this coupling can be. Perhaps the plot would have been more clearer if they was in log scale.}

Following this suggestion, we adopted a logarithmic scale for $y_\psi$ in Fig.\ 9.2.

\item {\it Incorporating $(g-2)_\mu$ within the model while maintaining perturbative couplings should give an upper limit on the new physics scale. What is the range of the masses and couplings that one can have in this model while incorporating DM and $(g-2)_\mu$? Comments on this matter greatly enhance the paper.}

As stated above eq.~(2.4) we restricted the coupling parameters $|\lambda_{ij}|\le 2$ to ensure perturbativity. While this choice may be overly conservative, we believe that a quantitative determination of the perturbativity bound on these couplings is beyond the scope of our work. In particular, the result will depend on the unknown scale and details of the UV completion of our model.

In addition, as discussed in section 9.1 and visible in Figure 9.4, an increased NP scale in conjunction with the solution of the $(g-2)_\mu$ anomaly leads to increased fine-tuning in the muon mass, leading to a naturalness requirement of the NP scale of around 2\,TeV. The mass range considered by us is therefore the one where the new particles are to be expected, if indeed our model solves the $(g-2)_\mu$ anomaly.

---

## Editorial Decision

published